# Generalisation Guarantees for Continual Learning with Orthogonal Gradient Descent

## Abstract

In Continual Learning settings, deep neural networks are prone to Catastrophic Forgetting. Orthogonal Gradient Descent was proposed to tackle the challenge. However, no theoretical guarantees have been proven yet. We present a theoretical framework to study Continual Learning algorithms in the Neural Tangent Kernel regime. This framework comprises closed form expression of the model through tasks and proxies for Transfer Learning, generalisation and tasks similarity. In this framework, we prove that OGD is robust to Catastrophic Forgetting then derive the first generalisation bound for SGD and OGD for Continual Learning. Finally, we study the limits of this framework in practice for OGD and highlight the importance of the Neural Tangent Kernel variation for Continual Learning with OGD.

## 1 Introduction

Continual Learning is a setting in which an agent is exposed to multiples tasks sequentially (Kirkpatrick et al., 2016). The core challenge lies in the ability of the agent to learn the new tasks while retaining the knowledge acquired from previous tasks. Too much plasticity (Nguyen et al., 2018) will lead to catastrophic forgetting, which means the degradation of the ability of the agent to perform the past tasks (McCloskey & Cohen 1989, Ratcliff 1990, Goodfellow et al. 2014). On the other hand, too much stability will hinder the agent from adapting to new tasks.

While there is a large literature on Continual Learning (Parisi et al., 2019), few works have addressed the problem from a theoretical perspective. Recently, Jacot et al. (2018) established the connection between overparameterized neural networks and kernel methods by introducing the Neural Tangent Kernel (NTK). They showed that at the infinite width limit, the kernel remains constant throughout training. Lee et al. (2019) also showed that, in the infinite width limit or Neural Tangent Kernel (NTK) regime, a network evolves as a linear model when trained on certain losses under gradient descent. In addition to these findings, recent works on the convergence of Stochastic Gradient Descent for overparameterized neural networks (Arora et al., 2019) have unlocked multiple mathematical tools to study the training dynamics of over-parameterized neural networks.

We leverage these theoretical findings in order to to propose a theoretical framework for Continual Learning in the NTK regime then prove convergence and generalisation properties for the algorithm Orthogonal Gradient Descent for Continual Learning (Farajtabar et al., 2019).

Our contributions are summarized as follows:

1. We present a theoretical framework to study Continual Learning algorithms in the Neural Tangent Kernel (NTK) regime. This framework frames Continual Learning as a recursive kernel regression and comprises proxies for Transfer Learning, generalisation, tasks similarity and Curriculum Learning. (Thm. 1, Lem. 1 and Thm. 3).

2. In this framework, we prove that OGD is robust to forgetting with respect to an arbitrary number of tasks under an infinite memory (Sec . 5, Thm. 2).

3. We prove the first generalisation bound for Continual Learning with SGD and OGD. We find that generalisation through tasks depends on a task similarity with respect to the NTK. (Sec. 5, Theorem 3)

4. We study the limits of this framework in practical settings, in which the Neural Tangent Kernel may vary. We find that the variation of the Neural Tangent Kernel impacts negatively

the robustness to Catastrophic Forgetting of OGD in non overparameterized benchmarks. (Sec. 6)

## 2 RELATED WORKS

Continual Learning addresses the Catastrophic Forgetting problem, which refers to the tendency of agents to "forget" the previous tasks they were trained on over the course of training. It's an active area of research, several heuristics were developed in order to characterise it (Ans & Rousset 1997, Ans & Rousset 2000, Goodfellow et al. 2014, French 1999, McCloskey & Cohen 1989, Robins 1995, Nguyen et al. 2019). Approaches to Continual Learning can be categorised into: regularization methods, memory based methods and dynamic architectural methods. We refer the reader to the survey by Parisi et al. (2019) for an extensive overview on the existing methods. The idea behind memory-based methods is to store data from previous tasks in a buffer of fixed size, which can then be reused during training on the current task (Chaudhry et al. 2019, Van de Ven & Tolias 2018). While dynamic architectural methods rely on growing architectures which keep the past knowledge fixed and store new knowledge in new components, such as new nodes or layers. (Lee et al. 2018, Schwarz et al. 2018) Finally, regularization methods regularize the objective in order to preserve the knowledge acquired from the previous tasks (Kirkpatrick et al. 2016, Aljundi et al. 2018, Farajtabar et al. 2019, Zenke et al. 2017).

While there is a large literature on the field, there is a limited number of theoretical works on Continual Learning. Alquier et al. (2017) define a compound regret for lifelong learning, as the regret with respect to the oracle who would have known the best common representation for all tasks in advance. Knoblauch et al. (2020) show that optimal Continual Learning algorithms generally solve an NP-HARD problem and will require perfect memory not to suffer from catastrophic forgetting. Benzing (2020) presents mathematical and empirical evidence that the two methods – Synaptic Intelligence and Memory Aware Synapses – approximate a rescaled version of the Fisher Information.

Continual Learning is not limited to Catastrophic Forgetting, but also closely related to Transfer Learning. A desirable property of a Continual Learning algorithm is to enable the agent to carry the acquired knowledge through his lifetime, and transfer it to solve new tasks. A new theoretical study of the phenomena was presented by Liu et al. (2019). They prove how the task similarity contributes to generalisation, when training with Stochastic Gradient Descent, in a two tasks setting and for over-parameterised two layer RELU neural networks.

The recent findings on the Neural Tangent Kernel (Jacot et al., 2018) and on the properties of overparameterized neural networks (Du et al. 2018, Arora et al. 2019) provide powerful tools to analyze their training dynamics. We build up on these advances to construct a theoretical framework for Continual Learning and study the generalisation properties of Orthogonal Gradient Descent.

## 3 PRELIMINARIES

**Notation**   We use bold-faced characters for vectors and matrices. We use $\|\cdot\|$ to denote the Euclidean norm of a vector or the spectral norm of a matrix, and $\|\cdot\|_F$ to denote the Frobenius norm of a matrix. We use $\langle\cdot,\cdot\rangle$ for the Euclidean dot product, and $\langle\cdot,\cdot\rangle_{\mathcal{H}}$ the dot product in the Hilbert space $\mathcal{H}$. We index the task ID by $\tau$. The $\leq$ operator if used with matrices, corresponds to the partial ordering over symmetric matrices. We denote $\mathbb{N}$ the set of natural numbers, $\mathbb{R}$ the space of real numbers and $\mathbb{N}^{\star}$ for the set $\mathbb{N} \smallsetminus \{0\}$. We use $\oplus$ to refer to the direct sum over Euclidean spaces.

### 3.1 CONTINUAL LEARNING

Continual Learning considers a series of tasks $\{\mathcal{T}_1, \mathcal{T}_2, \ldots\}$, where each task can be viewed as a separate supervised learning problem. Similarly to online learning, data from each task is revealed only once. The goal of Continual Learning is to model each task accurately with a single model. The challenge is to achieve a good performance on the new tasks, while retaining knowledge from the previous tasks (Nguyen et al., 2018).

We assume the data from each task $\mathcal{T}_\tau, \tau \in \mathbb{N}^{\star}$, is drawn from a distribution $\mathcal{D}_\tau$. Individual samples are denoted $(\mathbf{x}_{\tau,i}, y_{\tau,i})$, where $i \in [n_\tau]$. For a given task $\mathcal{T}_\tau$, the model is denoted $f_\tau$, we use the

superscript $(t)$ to indicate the training iteration $t \in \mathbb{N}$, while we use the superscript $\star$ to indicate the asymptotic convergence. For the regression case, given a ridge regularisation coefficient $\lambda \in \mathbb{R}^+$, for all $t \in \mathbb{N}$, we write the train loss for a task $\mathcal{T}_\tau$ as :

$$\mathcal{L}^\tau(\mathbf{w}_\tau(t)) = \sum_{i=1}^{n_\tau} (f_\tau^{(t)}(\mathbf{x}_{\tau,i}) - y_{\tau,i})^2 + \lambda \big\| \mathbf{w}_\tau(t) - \mathbf{w}_{\tau-1}^\star \big\|^2.$$

## 3.2 OGD for Continual Learning

Let $\mathcal{T}_T$ the current task, where $T \in \mathbb{N}^\star$. For all $i \in [n_T]$, let $\mathbf{v}_{T,i} = \nabla_{\mathbf{w}} f_{T-1}^\star(\mathbf{x}_{T-1,i})$, which is the Jacobian of task $\mathcal{T}_T$. We define $\mathbb{E}_\tau = \mathrm{vec}(\{\mathbf{v}_{\tau,i}, i \in [n_\tau]\})$, the subspace induced by the Jacobian. The idea behind OGD (Farajtabar et al., 2019) is to update the weights along the projection of the gradient on the orthogonal space induced by the Jacobians over the previous tasks $\mathbb{E}_1 \oplus \ldots \oplus \mathbb{E}_{\tau-1}$. The update rule at an iteration $t \in \mathbb{N}^\star$ for the task $\mathcal{T}_T$ is as follows :

$$\mathbf{w}_T(t+1) = \mathbf{w}_T(t) - \eta \Pi_{\mathbb{E}_{T-1}^\perp} \nabla_{\mathbf{w}} \mathcal{L}^T(\mathbf{w}_T(t)).$$

The intuition behind OGD is to "preserve the previously acquired knowledge by maintaining a space consisting of the gradient directions of the neural networks predictions on previous tasks" (Farajtabar et al., 2019). Throughout the paper, we only consider the OGD-GTL variant which stores the gradient with respect to the ground truth logit.

## 3.3 Neural Tangent Kernel

In their seminal paper, Jacot et al. (2018) established the connection between deep networks and kernel methods by introducing the Neural Tangent Kernel (NTK). They showed that at the infinite width limit, the kernel remains constant throughout training. Lee et al. (2019) also showed that a network evolves as a linear model in the infinite width limit when trained on certain losses under gradient descent.

Throughout our analysis, we make the assumption that the neural network is overparameterized, and consider the linear approximation of the neural network around its initialisation:

$$f^{(t)}(\mathbf{x}) \approx f^{(0)}(\mathbf{x}) + \nabla_{\mathbf{w}} f^{(0)}(\mathbf{x})^T (\mathbf{w}(t) - \mathbf{w}(0)).$$

## 4 Convergence - Continual Learning as a recursive Kernel Regression

In this section, we derive a closed form expression for the Continual Learning models through tasks. We find that Continual Learning models can be expressed with recursive kernel ridge regression across tasks. We also find a that the NTK of OGD is recursive with respect to the projection of its feature map on the tasks' spaces. The result is presented in Theorem 1, a stepping stone towards proving the generalisation bound for OGD in Sec. 5.

### 4.1 Convergence Theorem

**Theorem 1 (Continual Learning as a recursive Kernel Regression)**

*Given $\mathcal{T}_1, \ldots, \mathcal{T}_T$ a sequence of tasks. Fix a learning rate sequence $(\eta_\tau)_{\tau \in [T]}$ and a ridge regularisation coefficient $\lambda \in \mathbb{R}^+$. If, for all $\tau$, the learning rate satisfies $\eta_\tau < \frac{1}{\|\kappa_\tau(\mathbf{X}_\tau, \mathbf{X}_\tau) + \lambda \mathbf{I}\|}$, then for all $\tau$, $\mathbf{w}_\tau(t)$ converges linearly to a limit solution $\mathbf{w}_\tau^\star$ such that*

$$f_\tau^\star(\mathbf{x}) = f_{\tau-1}^\star(\mathbf{x}) + \kappa_\tau(\mathbf{x}, \mathbf{X}_\tau)^T (\kappa_\tau(\mathbf{X}_\tau, \mathbf{X}_\tau) + \lambda \mathbf{I})^{-1} \tilde{\mathbf{y}}_\tau,$$

*where*

$$\kappa_\tau(\mathbf{x}, \mathbf{x}') = \widetilde{\phi}_\tau(\mathbf{x})\widetilde{\phi}_\tau(\mathbf{x}')^T,$$
$$\tilde{\mathbf{y}}_\tau = \mathbf{y}_\tau - \mathbf{y}_{\tau-1\to\tau},$$
$$\mathbf{y}_{\tau-1\to\tau} = f_{\tau-1}^\star(\mathbf{X}_\tau),$$
$$\widetilde{\phi}_\tau(\mathbf{x}) = \begin{cases} \nabla_{\mathbf{w}} f_0^\star(\mathbf{x}) \in \mathbb{R}^d & \text{for SGD}, \\ \mathbf{T}_\tau \nabla_{\mathbf{w}} f_0^\star(\mathbf{x}) \in \mathbb{R}^{d-M_\tau} & \text{for OGD}. \end{cases}$$

*and $\{\mathbf{T}_\tau \in \mathbb{R}^{d-M_\tau,d}, \tau \in [T]\}$ are projection matrices from $\mathbb{R}^d$ to $(\bigoplus_{k=1}^{\tau} \mathbb{E}_k)^\perp$ and $M_\tau = dim(\bigoplus_{k=1}^{\tau} \mathbb{E}_k)$ .*

The theorem describes how the model $f_\tau^\star$ evolves across tasks. It is recursive because the learning is incremental. For a given task $\mathcal{T}_\tau$, $f_{\tau-1}^\star(\mathbf{x})$ is the knowledge acquired by the agent up to the task $\mathcal{T}_{\tau-1}$. At this stage, the model only fits the residual $\tilde{\mathbf{y}}_\tau = \mathbf{y}_\tau - \mathbf{y}_{\tau-1\to\tau}$, which complements the knowledge acquired through previous tasks. This residual is also a proxy for task similarity. If the tasks are identical, the residual is equal to zero. The knowledge increment is captured by the term: $\kappa_\tau(\mathbf{x}, \mathbf{X}_\tau)^T(\kappa_{\tau+1}(\mathbf{X}_\tau, \mathbf{X}_\tau) + \lambda\mathbf{I})^{-1}\tilde{\mathbf{y}}_\tau$. Finally, the task similarity is computed with respect to the most recent feature map $\widetilde{\phi}_\tau$, and $\kappa_\tau$ is the NTK with respect to the feature map $\widetilde{\phi}_\tau$.

**Remark 1** *The recursive relation from Theorem 1 can also be written as a linear combination of kernel regressors as follows:*

$$f_\tau^\star(\mathbf{x}) = \sum_{k=1}^\tau \tilde{f}_k^\star(\mathbf{x}),$$

*where*

$$\tilde{f}_k^\star(\mathbf{x}) = \kappa_k(\mathbf{x}, \mathbf{X}_k)^T(\kappa_k(\mathbf{X}_k, \mathbf{X}_k) + \lambda\mathbf{I})^{-1}\tilde{\mathbf{y}}_k.$$

## 4.2 DISTANCE FROM INITIALISATION THROUGH TASKS

As described in Sec. 4.1, $\tilde{\mathbf{y}}_\tau$ is a residual. It is equal to zero if the model $f_{\tau-1}^\star$ makes perfect predictions on the next task $\mathcal{T}_\tau$. The more the next task $\mathcal{T}_\tau$ is different, the further the neural network needs to move from its previous state in order to fit it. Corollary 1 tracks the distance from initialisation as a function of task similarity.

**Corollary 1** *For SGD, and for OGD under the additional assumption that $\{\mathbf{T}_\tau, \tau \in [T]\}$ are orthonormal,*

$$\left\| \mathbf{w}_{\tau+1}^\star - \mathbf{w}_\tau^\star \right\|^2 = \tilde{\mathbf{y}}_{\tau+1}^T(\kappa(\mathbf{X}_{\tau+1}, \mathbf{X}_{\tau+1}) + \lambda\mathbf{I})^{-1}\kappa(\mathbf{X}_{\tau+1}, \mathbf{X}_{\tau+1})(\kappa(\mathbf{X}_{\tau+1}, \mathbf{X}_{\tau+1}) + \lambda\mathbf{I})^{-1}\tilde{\mathbf{y}}_{\tau+1},$$

**Remark 2** *Corollary 1 can be applied to get a similar result to Theorem 3 by Liu et al. (2019). In this remark, we consider mostly their notations. Their theorem states that under some conditions, for 2-layer neural networks with a RELU activation function, with probability no less than $1 - \delta$ over random initialisation,*

$$\|\mathbf{W}(P) - \mathbf{W}(Q)\|_{\mathrm{F}} \le \sqrt{\tilde{\mathbf{y}}_{P\to Q}^T H_P^{\infty-1}\tilde{\mathbf{y}}_{P\to Q}} + \epsilon,$$

*where, in their work:*

$$\mathbf{y}_{P\to Q} = H_{PQ}^{\infty,T} H_P^{\infty-1}\mathbf{y}_P,$$
$$\tilde{\mathbf{y}}_{P\to Q} = \mathbf{y}_Q - \mathbf{y}_{P\to Q}.$$

*Note that $H_P^\infty$ is a Gram matrix, which also corresponds to the NTK of the neural network they consider. We see an analogy with our result, where we work directly with the NTK, with no assumptions on the neural network. One important observation is that, to our knowledge, since there are no guarantees for the invertibility of our Gram matrix, we add a ridge regularisation to work with a regularised matrix, which is then invertible. In our setting, by considering $\lambda \to 0$, and with the additional assumption of invertibility of $\mathbf{H}_{\tau,0}$, which is valid in the two-layer overparameterized RELU neural network considered in the setting of Liu et al. (2019), we can recover a similar approximation.*

## 5 OGD : LEARNING WITHOUT FORGETTING, PROVABLY

In this section, we study the generalisation properties of OGD building-up on Thm. 1. First, we prove that OGD is robust to catastrophic forgetting with respect to all previous tasks (Theorem 2). Then, we present the main generalisation theorem for OGD (Thm. 3). The theorem provides several insights on the relation between task similarity and generalisation. Finally, we present how the Rademacher complexity relates to task similarity across a large number of tasks (Lemma 1). The lemma states that the more dissimilar tasks are, the larger the class of functions explored by the neural network, with high probability. This result highlights the importance of the curriculum for Continual Learning.

### 5.1 MEMORISATION PROPERTY OF OGD

The key to obtaining tight generalisation bounds for OGD is Theorem 2.

**Theorem 2 (No-forgetting Continual Learning with OGD)** *Given a task $\mathcal{T}_\tau$, for all $\mathbf{x}_{k,i} \in D_k$, a sample from the training data of a previous task $\mathcal{T}_k$, given that the Jacobian of $\mathbf{x}_{k,i}$ belongs to OGD's memory, it holds that:*

$$f_\tau^\star(\mathbf{x}_{k,i}) = f_k^\star(\mathbf{x}_{k,i}).$$

As motivated by Farajtabar et al. (2019), the orthogonality of the gradient updates aims to preserve the acquired knowledge, by not altering the weights along relevant dimensions when learning new tasks. Theorem 2 implies that, given an infinite memory, the training error on all samples from the previous tasks is unchanged, when training with OGD.

### 5.2 GENERALISATION PROPERTIES OF SGD AND OGD

Now, we state the main generalisation theorem for OGD, which provides generalisation bounds on the data from all the tasks, for SGD and OGD.

**Theorem 3 (Generalisation of SGD and OGD for Continual Learning)**

*Let $\{\mathcal{T}_1, \ldots \mathcal{T}_T\}$ be a sequence of tasks. Let be $\{\mathcal{D}_1, \ldots, \mathcal{D}_T\}$ the respective distributions over $\mathbb{R}^d \times \{-1, 1\}$. Let $\{(\mathbf{x}_{\tau,i}, y_{\tau,i}), i \in [n_t], \tau \in [T]\}$ be i.i.d. samples from $\mathcal{D}_\tau, \tau \in [T]$. Denote $\mathbf{X}_\tau = (\mathbf{x}_{\tau,1}, \ldots, \mathbf{x}_{\tau,n_\tau})$, $\mathbf{y}_\tau = (y_{\tau,1}, \ldots, y_{\tau,n_\tau})$. Consider the kernel ridge regression solution $f_T^\star$, then, for any loss function $\ell : \mathbb{R} \times \mathbb{R} \to [0, c]$ that is c-Lipschitz in the first argument, with probability at least $1 - \delta$,*

$$\mathcal{L}_{D_\tau}(f_T^\star) \leq \begin{cases} \frac{\lambda^2}{n_\tau} \tilde{\mathbf{y}}_\tau (\kappa_\tau(\mathbf{X}_\tau, \mathbf{X}_\tau) + \lambda\mathbf{I})^{-1} \tilde{\mathbf{y}}_\tau + R_T + 3c\sqrt{\frac{\log(2/\delta)}{2n_T}}, & \text{for OGD}, \tau \in [1, T], \\ \frac{\lambda^2}{n_\tau} \tilde{\mathbf{y}}_\tau (\kappa_\tau(\mathbf{X}_\tau, \mathbf{X}_\tau) + \lambda\mathbf{I})^{-1} \tilde{\mathbf{y}}_\tau + R_T + 3c\sqrt{\frac{\log(2/\delta)}{2n_T}}, & \text{for SGD}, \tau = T, \\ \frac{\lambda^2}{n_\tau} \tilde{\mathbf{y}}_\tau^T (k_\tau(\mathbf{X}_\tau, \mathbf{X}_\tau) + \lambda\mathbf{I})^{-1} \tilde{\mathbf{y}}_\tau + \frac{1}{n_T} \sum_{k=\tau+1}^T H_{k,\tau} + R_T + 3c\sqrt{\frac{\log(2/\delta)}{2n_T}}, & \text{for SGD}, \tau < T. \end{cases}$$

*where*

$$\tilde{\mathbf{y}}_\tau = \mathbf{y}_\tau - f_{\tau-1}^\star(\mathbf{X}_\tau),$$

$$R_T = \sum_{\tau=1}^T \sqrt{\frac{\mathrm{tr}(\kappa_\tau(\mathbf{X}_\tau, \mathbf{X}_\tau))}{n_\tau^2} \tilde{\mathbf{y}}_\tau^T (\kappa_\tau(\mathbf{X}_\tau, \mathbf{X}_\tau) + \lambda\mathbf{I})^{-1} \tilde{\mathbf{y}}_\tau},$$

$$H_{k,\tau} = \tilde{\mathbf{y}}_k^T (k_k(\mathbf{X}_k, \mathbf{X}_k) + \lambda\mathbf{I})^{-1} k_k(\mathbf{X}_k, \mathbf{X}_\tau) k_k(\mathbf{X}_\tau, \mathbf{X}_k)(k_k(\mathbf{X}_k, \mathbf{X}_k) + \lambda\mathbf{I})^{-1} \tilde{\mathbf{y}}_k.$$

Theorem 3 shows that the generalisation bound for OGD is tighter than SGD. This results from Thm. 2, which states that OGD is robust to Catastrophic Forgetting. The bound is looser for SGD over the past tasks through the forgetting term that appears.

The bound of Theorem 3 comprises the following main terms :

- $\frac{\lambda^2}{n_\tau} \tilde{\mathbf{y}}_\tau (\kappa_\tau(\mathbf{X}_\tau, \mathbf{X}_\tau) + \lambda\mathbf{I})^{-1} \tilde{\mathbf{y}}_\tau$ : this term is due to the regularisation and describes that the empirical loss is not equal to zero due to the regularisation term. This term tends to zero in case there is no regularisation. For interpretation purposes, in App. D.2.7, we show that the empirical loss tends to zero in the no-regularisation case.

- $R_T$ captures the impact of task similarity on generalisation which we discuss in Sec. 5.3.
- $\sum_{k=\tau+1}^{T} H_{k,\tau}$ is a residual term that appears for SGD only. It is due to the catastrophic forgetting that occurs with SGD. It also depends on the tasks similarity.

These bounds share some similarities with the bounds derived by Arora et al. (2019), Liu et al. (2019) and Hu et al. (2019), where in these works, the bounds were derived for supervised learning settings, and in some cases for two-layer RELU neural networks. Similarly, the bounds depend on the Gram matrix of the data, with the feature map corresponding to the NTK.

### 5.3 THE IMPACT OF TASK SIMILARITY ON GENERALISATION

Now, we state Lemma 1, which tracks the Rademacher complexity through tasks.

**Lemma 1** *Keeping the same notations and setting as Theorem 3, the Rademacher Complexity can be bounded as follows:*

$$\hat{\mathcal{R}}(\mathcal{F}_T) \leq \sum_{\tau=1}^{T} \mathcal{O}\left(\sqrt{\frac{\mathrm{tr}(\kappa_\tau(\mathbf{X}_\tau, \mathbf{X}_\tau))}{n_\tau^2} \tilde{\mathbf{y}}_\tau^T (\kappa_\tau(\mathbf{X}_\tau, \mathbf{X}_\tau) + \lambda \mathbf{I})^{-1} \tilde{\mathbf{y}}_\tau}\right);$$

*where $\mathcal{F}_T$ is the function class covered by the model up to the task $\mathcal{T}_\tau$*

The right hand side term $R_T$ in the upper bound of the generalisation theorem follows directly from Lemma 1, and it draws a **complexity measure for Continual Learning**. It states that the upper bound on the Rademacher complexity increases when the tasks are dissimilar. We define the *NTK task dissimilarity* between two subsequent tasks $\mathcal{T}_{\tau-1}$ and $\mathcal{T}_\tau$ as $\bar{S}_{\tau-1\to\tau} = \tilde{\mathbf{y}}_\tau^T(\kappa_\tau(\mathbf{X}_\tau, \mathbf{X}_\tau) + \lambda \mathbf{I})^{-1}\tilde{\mathbf{y}}_\tau$. This dissimilarity is a generalisation of the term that appears in the upper bound of Thm. 2 by Liu et al. 2019. The knowledge from the previous tasks is encoded in the kernel $\kappa_\tau$, through the feature map $\phi_\tau$. As an edge case, if two successive tasks are identical, $\bar{S}_{\tau-1\to\tau} = 0$ and the upper bound does not increase.

**Implications for Curriculum Learning** We also observe that the upper bound depends on the task ordering, which may provide a theoretical explanation on the importance of learning with a curriculum (Bengio et al. (2009)). In the following, we present an edge case which provided an intuition on how the bound captures the importance of the order. Consider two dissimilar tasks $\mathcal{T}_1$ and $\mathcal{T}_2$. A sequence of tasks alternating between $\mathcal{T}_1$ and $\mathcal{T}_2$ will lead to a large upper bound, as explained in the first paragraph, while a sequence of tasks concatenating two sequences of $\mathcal{T}_1$ then $\mathcal{T}_2$ will lead to a lower upper bound.

### 6 THE IMPACT OF THE NTK VARIATION ON OGD

In the previous section, we demonstrated that under the NTK regime and an infinite memory, OGD is provably robust to catastrophic forgetting. In practical settings, these two assumptions do not hold, therefore, in this section, we study the limits of the NTK regime. We present OGD+, a variant of OGD we use to study the importance of the NTK variation for Continual Learning in practice.

In order to decouple the NTK variation phenomena in our experiments, we propose then study the OGD+ algorithm, which is designed to be more robust to the NTK variation, through updating its orthonormal basis with respect to all the tasks **at the end of each task**, as opposed to OGD.

Algorithm 1 presents the OGD+ algorithm, we highlight the differences with OGD in red. The main difference is that OGD+ stores the feature maps with respect to the samples from previous tasks, in addition to the feature maps with respect to the samples from the current task, as opposed to OGD. This small change is motivated by the variation of the NTK in practice. In order to compute the feature maps with respect to the previous samples, OGD+ saves these samples in a dedicated memory, we call this storage the *samples memory*. This memory comes in addition to the orthonormal *feature maps memory*. The only role of the *samples memory* is to compute the updated feature maps at the beginning of each task.

natbib

---

**Algorithm 1:** OGD+ for Continual Learning

> **Input** : A task sequence $\mathcal{T}_1, \mathcal{T}_2, \ldots$, learning rate $\eta$
>
>      1. Initialize $S_J \leftarrow \{\}$ ; $S_D \leftarrow \{\}$; $\mathbf{w} \leftarrow \mathbf{w}_0$
>
>      2. **for** *Task ID* $\tau = 1, 2, 3, \ldots$ **do**
>
> | **repeat**
> |    | $\mathbf{g} \leftarrow$ Stochastic Batch Gradient for $\mathcal{T}_\tau$ at $\mathbf{w}$;
> |    | $\tilde{\mathbf{g}} = \mathbf{g} - \sum_{\mathbf{V} \in \mathcal{S}_J} \mathrm{proj}_{\mathbf{V}}(\mathbf{g})$;
> |    | $\mathbf{w} \leftarrow \mathbf{w} - \eta \tilde{\mathbf{g}}$
> | **until** *convergence*;
> | Sample $\mathcal{S} \subset \mathcal{S}_D$;
> | **for** $(\mathbf{x}, y) \in \mathcal{D}_\tau \bigcup \mathcal{S}$ *and* $k \in [1, c]$ *s.t.* $y_k = 1$ **do**
> |    | $\mathbf{u} \leftarrow \nabla f_\tau(\mathbf{x}; \mathbf{w}) - \sum_{\mathbf{V} \in \mathcal{S}_J} \mathrm{proj}_{\mathbf{V}}(\nabla f_\tau(\mathbf{x}; \mathbf{w}))$ $\mathcal{S}_J \leftarrow \mathcal{S}_J \bigcup \{\mathbf{u}\}$
> | **end**
> | Sample $\mathcal{D} \subset \mathcal{D}_\tau$ ;
> | Update $\mathcal{S}_D \leftarrow \mathcal{S}_D \bigcup \mathcal{D}$
>      **end**

---

## 7 EXPERIMENTS

We study the validity Theorem 2 which states that under the given conditions, OGD is perfectly robust to Catastrophic Forgetting (Sec. 7.1). We find that the Catastrophic Forgetting of OGD decreases with over-parameterization. Then, we perform an ablation study using OGD+ in order to study the applicability and limits of the constant Jacobian assumption in practice (Sec. 7.2). We find that the assumption holds for some benchmarks, and that updating the Jacobian (OGD+) is critical for the robustness of OGD in the non-overparameterized benchmarks. Finally, we present a broader picture of the performance of OGD+ against standard Continual Learning baselines (Sec. 7.3)

**Benchmarks** We use the standard benchmarks similarly to Goodfellow et al. (2014) and Chaudhry et al. (2019). Permuted MNIST (Goodfellow et al., 2014) consists of a series of MNIST supervised learning tasks, where the pixels of each task are permuted with respect to a fixed permutation. Rotated MNIST (Farajtabar et al., 2019) consists of a series of MNIST classification tasks, where the images are rotated with respect to a fixed angle, monotonically. We increment the rotation angle by 5 degrees at each new task. Split CIFAR-100 (Chaudhry et al., 2019) is constructed by splitting the original CIFAR-100 dataset (Krizhevsky, 2009) into 20 disjoint subsets, where each subset is formed by sampling without replacement 5 classes out of 100. In order to assess the robustness to catastrophic forgetting over long tasks sequences, we increase the length of the tasks streams from 5 to 15 for the MNIST benchmarks, and consider all the 20 tasks for the Split CIFAR-100 benchmark. We also use the CUB200 benchmark (Jung et al., 2020) which contains 200 classes, split into 10 tasks. This benchmark is more overparameterized than the other benchmarks.

**Evaluation** Denoting $a_{T,\tau}$ the accuracy of the model on the task $\mathcal{T}_\tau$ after being trained on the task $\mathcal{T}_T$, we track two metrics introduced by Chaudhry et al. (2019) and Lopez-Paz & Ranzato (2017) :

AVERAGE ACCURACY ($A_T$) is the average accuracy after the model has been trained on the task $\mathcal{T}_T$ :

$$A_T = \frac{1}{T} \sum_{\tau=1}^{T} a_{T,\tau}.$$

AVERAGE FORGETTING ($F_T$) is the average forgetting after the model has been trained on the task $\mathcal{T}_T$ :

$$F_T = \frac{1}{T-1} \sum_{\tau=1}^{T-1} \max_{t \in \{1, \ldots, T-1\}} (a_{t,\tau} - a_{T,\tau}).$$

OVER-PARAMETERIZATION    We track over-parameterization of a given benchmark as the number of trainable parameters over the average number of samples per task (Arora et al., 2019) :

$$O = \frac{p}{\sum_{\tau=1}^{T} n_\tau / T}.$$

**Architectures**    For the MNIST benchmarks, we use the same neural network architectures as Farajtabar et al. (2019). However, since the OGD algorithm doesn't scale to large neural networks due to memory limits, we considered smaller scale neural networks for the CIFAR100 and CUB200 benchmarks. For CIFAR100, we use the LeNet architecture (Lecun et al., 1998). For CUB200, we keep the pretrained AlexNet similarly to Jung et al. (2020), then freeze the features layers and replace the default classifier with a smaller neural network (App. F.1.1).

### 7.1    ABLATION STUDY : FOR OGD, CATASTROPHIC FORGETTING DECREASES WITH OVERPARAMETERIZATION (THM. 2)

Theorem 2 states that in the NTK regime, given an infinite memory, the train error of OGD is unchanged. We study the impact of overparameterization on Catastrophic Forgetting of OGD through an ablation study on the number of parameters of the model.

**Experiment**    We train a model on the full task sequence on the MNIST and CIFAR100 benchmarks, then measure the variation of the train accuracy of the memorised samples from the first task. The reason we consider the samples in the memory is the applicability of Thm. 2 to these samples only. Fixing the datasets sizes, our proxy for overparameterization is the hidden size, which we vary.

**Results**    Figures 1 and 2 show that the train error variation decreases with overparameterization for OGD, on the MNIST and CIFAR100 benchmarks. This result concurs with Thm. 2.

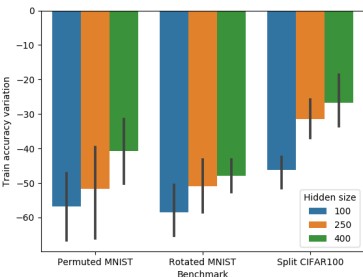
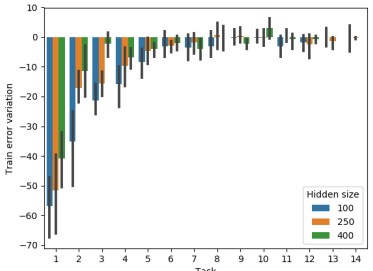

Figure 1: The variation of the train accuracy on the memorised samples *from the first task* as a function of overparameterization (higher is better). The forgetting decreases with overparameterization, as stated in Theorem 2.

Figure 2: The variation of the train accuracy on the memorised samples *from each task*, after the model was trained on all tasks in sequence (higher is better). We vary the hidden size as a proxy for overparameterization.

### 7.2    ABLATION STUDY : FOR NON OVER-PARAMETERIZED BENCHMARKS, UPDATING THE JACOBIAN IS CRITICAL TO OGD'S ROBUSTNESS (THM. 2)

Our analysis relies on the over-parameterization assumption, which implies that the Jacobian is constant through tasks. Theorem 2 follows from this property. However, in practice, this assumption may not hold and forgetting is observed for OGD. We study the impact of the variation of the Jacobian in practice on OGD.

In order to measure this impact, we perform an ablation study on OGD, taking into account the Jacobian's variation (OGD+) or not (OGD). OGD+ takes into account the Jacobian's variation by updating all the stored Jacobians at the end of each task.

**Experiment**    We measure the average forgetting (Chaudhry et al., 2019) without (OGD) or with (OGD+) accounting for the Jacobian's variation, on the MNIST, CIFAR100 and CUB200 benchmarks.

**Results** Table 1 shows that for the Rotated MNIST and Permuted MNIST benchmarks, which are the least overparameterized, OGD+ is more robust to Catastrophic Forgetting than OGD. This improvement follows from the variation of the Jacobian in these benchmarks and that OGD+ accounts for it. While on CIFAR100 and CUB200, the most over-parameterized benchmark, the robustness of OGD and OGD+ is equivalent. For these benchmarks, since the variation of the Jacobian is smaller, due to overparameterization, OGD+ is equivalent to OGD.

This result confirms our initial hypothesis that the Jacobian's variation in non-overparameterized settings such as MNIST is an important reason the OGD algorithm is prone to Catastrophic Forgetting. This result also highlights the importance of developing a theoretical framework for the non-overparameterized setting, in order to capture the properties of OGD outside the NTK regime.

| Dataset | Permuted MNIST | Rotated MNIST | Split CIFAR100 | CUB200 |
|---|---|---|---|---|
| **Overparameterization** | 2 | 2 | 28 | 1166 |
| **OGD** | -9.71 (±0.95) | -17.77 (±0.35) | -20.86 (±1.41) | -12.99 (±0.92) |
| **OGD+** | **-5.98 (±0.36)** | **-8.57 (±0.53)** | -21.59 (±1.27) | -12.89 (±0.51) |

Table 1: The average forgetting of with and without accounting for the Jacobian's variation, on the MNIST, CIFAR and CUB200 datasets (lower is better). A higher over-parameterization coefficient implies that the constant Jacobian assumption is more likely to hold.

### 7.3 BENCHMARKING OGD+ AGAINST OTHER CONTINUAL LEARNING BASELINES

In order to provide a broader picture on the robustness of OGD+ to Catastrophic Forgetting, we benchmark it against standard Continual Learning baselines.

**Results** Table 2 shows that, on the Permuted MNIST and Rotated MNIST benchmarks, which are the least overparameterized, OGD+ draws an improvement over OGD and is competitive with other Continual Learning methods. On the most overparameterized benchmarks, CIFAR100 and CUB200, OGD+ is not competitive.

| | Permuted MNIST | Rotated MNIST | Split CIFAR100 | CUB200 |
|---|---|---|---|---|
| **Naive SGD** | 76.31 (±1.89) | 71.06 (±0.41) | 50.77 (±3.99) | 53.93 (±0.86) |
| **EWC** | 80.85 (±1.14) | 80.96 (±0.42) | 56.82 (±1.75) | 62.15 (±0.37) |
| **SI** | 86.69 (±0.4) | 75.33 (±0.55) | 66.66 (±2.07) | **63.17 (±0.42)** |
| **MAS** | 85.96 (±0.72) | 80.55 (±0.46) | 66.33 (±1.13) | 61.43 (±0.68) |
| **Stable SGD** | 78.17 (±0.76) | **88.92 (±0.19)** | **72.86 (±0.9)** | 58.79 (±0.36) |
| **OGD** | 85.0 (±0.86) | 79.17 (±0.33) | 61.82 (±1.24) | 57.89 (±0.89) |
| **OGD+** | **88.65 (±0.38)** | 87.73 (±0.5) | 61.11 (±1.31) | 57.99 (±0.52) |

Table 2: The average accuracy of several methods on the MNIST, CIFAR and CUB200 datasets.

## 8 CONCLUSION

We presented a theoretical framework for Continual Learning algorithms in the NTK regime, then leveraged the framework to study the convergence and generalisation properties of the SGD and OGD algorithms. We also assessed our theoretical results through experiments and highlighted the applicability of the framework in practice. Our analysis highlights multiple connections to neighbouring fields such as Transfer Learning and Curriculum Learning. Extending the analysis to other Continual Learning algorithms is a promising future direction, which would provide a better understanding of the various properties of these algorithms and how they could be improved. Additionally, our experiments highlight the limits of the applicability of the framework to non-overparameterized settings. In order to gain a better understanding of the properties of the OGD algorithm in this setting, extending the framework beyond the over-parameterization assumption is an important direction. We hope this work provides new keys to investigate these directions.

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

# A COMPLEMENTARY DISCUSSION

## A.1 NTK VARIATION - THE IMPORTANCE OF ORTHOGONALITY FOR THE OGD, OGD+ AND A-GEM ALGORITHMS

In the main article, we focused only on the SGD, OGD and OGD+ algorithms. In this section, we highlight a connection of these algorithms to the A-GEM algorithm. All these algorithms perform orthogonal projections during the weight update step, however these projections differ in the following ways :

- the span of the space the updates are orthogonal to.
- the rate at which the projection space is updated.

Additionally, the A-GEM algorithm present the additional property of allowing Positive Backward Transfer by design. Positive Backward Transfer is a desirable property in the sense that the generalisation on a past task increases while training on the current task.

**Similarities among the algorithms**    Table 3 presents an overview of the connections between the algorithm. In practice, the models are not trained in the NTK regime. Since the feature map changes through training, the orthogonality constraint of OGD becomes less relevant and may harm the learning. On the other hand, while A-GEM performs a projection on a smaller subspace which may not protect all the subspaces, the subspace is updated as it is computed at each training step, the updates are therefore expected to be more relevant. Finally OGD+ lies at the intersection of both, while it doesn't update the feature maps at each gradient step, the projection constraint is with respect to a larger space.

| | Projection Space | | |
|---|---|---|---|
| Algorithm | Span | Update rate | Backward Transfer |
| SGD | None | None | No |
| OGD | Full | Task | Unclear |
| OGD+ | Full | Task | Unclear |
| A-GEM | Sample | Gradient step | Yes |

Table 3: Overview of the properties of the SGD, OGD, OGD+, A-GEM and A-GEM-NT algorithms.

**On the theoretical connection between OGD and A-GEM**    The A-GEM algorithm (Chaudhry et al., 2019) is a state of the art Continual Learning algorithm on the standard benchmarks. The idea behind the algorithm is to perform a gradient descent if an estimate of the loss over the previous tasks increases or is unchanged. Otherwise the gradient is projected orthogonally to the gradient over the loss estimate over the previous tasks.

We find that OGD draws a upper bound in comparison to A-GEM-NT in terms of generalisation error, as stated in Proposition 1.

**Proposition 1** *In the NTK regime, OGD implies A-GEM with no Positive Backward Transfer.*

The proof is presented in App. E.1.

**Experiments**    The extended experiments on the importance of the NTK variation for Continual Learning (Sec. F.3), show that A-GEM outperforms OGD and OGD+ on most benchmarks. A probable reason behind this difference is the rate of update of the feature map for the A-GEM algorithm, even though it spans a smaller space than OGD and OGD+.

## A.2 EXPRESSIONS OVERVIEW

For convenience, we present in Table 4 an overview of the notions that were mentioned in the main article, with their respective mathematical expressions.

| Name | Expression |
|------|------------|
| Residual | $\tilde{\mathbf{y}}_\tau = \mathbf{y}_\tau - \mathbf{y}_{\tau-1\to\tau}$ |
| Knowledge increment | $\tilde{f}_\tau^\star(\mathbf{x}) = \kappa_\tau(\mathbf{x}, \mathbf{X}_\tau)^T(\kappa_{\tau+1}(\mathbf{X}_\tau, \mathbf{X}_\tau) + \lambda\mathbf{I})^{-1}\tilde{\mathbf{y}}_\tau$ |
| Projection matrix | $\mathbf{T}_\tau$ |
| Feature map | $\phi(\mathbf{x})$ |
| Effective feature map | $\widetilde{\phi}_\tau(\mathbf{x})$ |
| Effective kernel | $\kappa_\tau(\mathbf{x}, \mathbf{x}')$ |
| Complexity measure | $\sum_{\tau=1}^T \sqrt{\frac{\mathrm{tr}(\kappa_\tau(\mathbf{X}_\tau,\mathbf{X}_\tau))}{n_\tau^2}\tilde{\mathbf{y}}_\tau^T(\kappa_\tau(\mathbf{X}_\tau, \mathbf{X}_\tau) + \lambda\mathbf{I})^{-1}\tilde{\mathbf{y}}_\tau}$ |
| NTK task dissimilarity | $\bar{S}_{\tau-1\to\tau} = \tilde{\mathbf{y}}_\tau^T(\kappa_\tau(\mathbf{X}_\tau, \mathbf{X}_\tau) + \lambda\mathbf{I})^{-1}\tilde{\mathbf{y}}_\tau$ |

Table 4: Overview of the notions presented in the main manuscript with their respective notations.

# B  PROOFS OVERVIEW

## B.1  CONVERGENCE :

**Theorem 1 : Convergence of SGD and OGD for Continual Learning**   We prove Theorem 1 by induction. We rewrite the loss function as a regression on the residual $\tilde{\mathbf{y}}_\tau$ instead of $\mathbf{y}_\tau$. Then, we rewrite the optimisation objective as an unconstrained strongly convex optimisation problem. Finally, we compute the unique solution in a closed form. The full proof is presented in App. C.1.

**Remark 1 : Continual Learning as a recursive Kernel Regression**   The remark follows directly from the recursive form of Theorem 1.

**Corollary 1 : Distance from initialisation through tasks**   The proof follows immediately from the proof of Theorem 1, in which we compute a closed form of the weights variation. It is presented in App. C.2.

## B.2  GENERALISATION :

**Theorem 2 : No-forgetting Continual Learning with OGD**   The proof relies on the orthogonality property of the OGD update, which implies that the subspaces spanned by the samples in the memory would not have any changes incurred. The proof is presented in App . D.1

**Theorem 3 : Generalisation of SGD and OGD for Continual Learning**   The proof or Theorem 3 is presented in App. D.2. The proof follows the following structure :

- Bounding the Rademacher complexity (App. D.2.3) : First, we state the technical Lemma 3 that upper bounds the Rademacher complexity of the function class that corresponds to the set of linear combinations of kernel regressors. Then we apply the lemma to the function class obtained through Theorem 1.
- Bounding the empirical loss for SGD (App. D.2.5) : the error is the same as OGD on the last task. While on the previous tasks, Catastrophic Forgetting can be incurred which implies the appearance of a residual term that corresponds to the forgetting.
- Bounding the empirical loss for OGD (App. D.2.4) : The proof techniques are very similar to the SGD case, however we leverage Theorem 2 in order to derive tighter bounds due to the absence of Catastrophic Forgetting.
- Wrap-up (App. D.2.6) : Finally, we wrap-up all the lemmas to derive the bound of Theorem 3.

**Lemma 1 - Implications for Curriculum Learning**   This results follows from the technical Lemma 3, which upper bounds the Rademacher complexity of the function class that corresponds to the set of linear combinations of kernel regressors.

## B.3  THE IMPORTANCE OF THE NTK VARIATION FOR CONTINUAL LEARNING :

**Proposition 1 - OGD implies A-GEM-NT**   The proof relies mainly on Theorem 2 stating the robustness of OGD to Catastrophic Forgetting. We derive the A-GEM update then apply Theorem 2, which leads to the implication. The full proof is presented in App. D.2.3.

## C  MISSING PROOFS OF SECTION 4 - CONVERGENCE

### C.1  PROOF OF THEOREM 1

**Stochastic Gradient Descent**    We start by proving the Stochastic Gradient Descent (SGD) case of Theorem 1.

**Proof**

We prove the Theorem 1 by induction.

Our induction hypothesis $H_\tau$ is the following : $\mathcal{H}_\tau$ : For all $k \leq \tau$, Theorem 1 holds.

First, we prove that $\mathcal{H}_1$ holds.

The proof is straightforward. For the first task, since there were no previous tasks, OGD on this task is the same as SGD.

Therefore, it is equivalent to minimising the following objective :

$$\arg\min_{\mathbf{w}\in\mathbb{R}^d} \left\| f_0(\mathbf{X}_1) + \phi(\mathbf{X}_1)^T(\mathbf{w} - \mathbf{w}_0^\star) - \mathbf{y}_1 \right\|_2^2 + \frac{\lambda}{2}\|\mathbf{w} - \mathbf{w}_0^\star\|^2 \tag{1}$$

where $\phi(\mathbf{x}) = \nabla_{\mathbf{w}_0^\star} f_0^\star(\mathbf{x})$.

We replace into the objective with $\tilde{\mathbf{y}}_1$ applying $\tilde{\mathbf{y}}_1 = \mathbf{y}_1 - f_0(\mathbf{X}_1)$ :

$$\arg\min_{\mathbf{w}\in\mathbb{R}^d} \left\| \phi(\mathbf{X}_1)^T(\mathbf{w} - \mathbf{w}_0^\star) - \tilde{\mathbf{y}}_1 \right\|_2^2 + \frac{\lambda}{2}\|\mathbf{w} - \mathbf{w}_0^\star\|^2 \tag{2}$$

The objective is quadratic and the Hessian is positive definite, therefore the minimum exists and is unique :

$$\mathbf{w}_1^\star - \mathbf{w}_0^\star = \phi(\mathbf{X}_1)(\phi(\mathbf{X}_1)^T\phi(\mathbf{X}_1) + \lambda\mathbf{I})^{-1}\tilde{\mathbf{y}}_1 \tag{3}$$

Under the NTK regime assumption :

$$f_1^\star(\mathbf{x}) = f_0^\star(\mathbf{x}) + \nabla_{\mathbf{w}} f_0^\star(\mathbf{x})^T(\mathbf{w}_1^\star - \mathbf{w}_0^\star) \tag{4}$$

Then, by replacing into $\mathbf{w}_1^\star - \mathbf{w}_0^\star$ :

$$f_1^\star(\mathbf{x}) = f_0^\star(\mathbf{x}) + \nabla_{\mathbf{w}} f_0^\star(\mathbf{x})^T\phi(\mathbf{X}_1)(\phi(\mathbf{X}_1)^T\phi(\mathbf{X}_1) + \lambda\mathbf{I})^{-1}\tilde{\mathbf{y}}_1 \tag{5}$$

$$f_1^\star(\mathbf{x}) = f_0^\star(\mathbf{x}) + \kappa_1(\mathbf{x}, \mathbf{X}_1)(\kappa_1(\mathbf{X}_1, \mathbf{X}_1) + \lambda\mathbf{I})^{-1}\tilde{\mathbf{y}}_1 \tag{6}$$

Finally :

$$f_1^\star(\mathbf{x}) - f_0^\star(\mathbf{x}) = \kappa_1(\mathbf{x}, \mathbf{X}_1)(\kappa_1(\mathbf{X}_1, \mathbf{X}_1) + \lambda\mathbf{I})^{-1}\tilde{\mathbf{y}}_1 \tag{7}$$

Which completes the proof of $\mathcal{H}_1$.

Let $\tau \in \mathbb{N}^\star$, we assume that $\mathcal{H}_\tau$ is true, then we show that $\mathcal{H}_{\tau+1}$ is true.

On the task $\mathcal{T}_{\tau+1}$, we can write the loss $\mathcal{L}^{\tau+1}$ as :

$$\mathcal{L}^{\tau+1}(\mathbf{w}) = \left\| \phi_\tau(\mathbf{X}_{\tau+1})^T(\mathbf{w} - \mathbf{w}_\tau^\star) - \tilde{\mathbf{y}}_{\tau+1} \right\|_2^2 + \frac{\lambda}{2}\|\mathbf{w} - \mathbf{w}_\tau^\star\|^2 \tag{8}$$

We recall that the optimisation problem on the task $\mathcal{T}_{\tau+1}$ is :

$$\arg\min_{\mathbf{w}\in\mathbb{R}^d} \left\| \phi_\tau(\mathbf{X}_{\tau+1})^T(\mathbf{w} - \mathbf{w}_\tau^\star) - \tilde{\mathbf{y}}_{\tau+1} \right\|_2^2 + \frac{\lambda}{2}\|\mathbf{w} - \mathbf{w}_\tau^\star\|^2 \tag{9}$$

The optimisation objective is quadratic, unconstrained, with a positive definite hessian. Therefore, an optimum exists and is unique :

$$\mathbf{w}_{\tau+1}^\star - \mathbf{w}_\tau^\star = \phi(\mathbf{X}_{\tau+1})(\phi(\mathbf{X}_{\tau+1})^T\phi(\mathbf{X}_{\tau+1}) + \lambda\mathbf{I})^{-1}\tilde{\mathbf{y}}_{\tau+1} \tag{10}$$

We define the kernel $\kappa_{\tau+1} : \mathbb{R}^d \times \mathbb{R}^d \to \mathbb{R}$ as :

$$\kappa_{\tau+1}(\mathbf{x}, \mathbf{x}') = \phi_\tau(\mathbf{x})^T \phi_\tau(\mathbf{x}') \quad \text{for all } \mathbf{x}, \mathbf{x}' \in \mathbb{R}^d \tag{11}$$

Finally, we recover a closed form expression for $f_{\tau+1}^\star$ :

First, we use the induction hypothesis $\mathcal{H}_\tau$ :

$$f_{\tau+1}^\star(\mathbf{x}) = f_\tau^\star(\mathbf{x}) + \langle \nabla_{\mathbf{w}} f_\tau^\star(\mathbf{x}), \mathbf{w}_{\tau+1}^\star - \mathbf{w}_\tau^\star \rangle \tag{12}$$

$$= f_\tau^\star(\mathbf{x}) + \phi_\tau(\mathbf{x})\phi(\mathbf{X}_{\tau+1})(\kappa_{\tau+1}(\mathbf{X}_{\tau+1}, \mathbf{X}_{\tau+1}) + \lambda\mathbf{I})^{-1}\tilde{\mathbf{y}}_{\tau+1} \tag{13}$$

$$= f_\tau^\star(\mathbf{x}) + \kappa_{\tau+1}(\mathbf{x}, \mathbf{X}_{\tau+1})(\kappa_{\tau+1}(\mathbf{X}_{\tau+1}, \mathbf{X}_{\tau+1}) + \lambda\mathbf{I})^{-1}\tilde{\mathbf{y}}_{\tau+1} \tag{14}$$

At this stage, we have proven $\mathcal{H}_{t+1}$.

$$\tag{15}$$

**Orthogonal Gradient Descent**   Now, we prove the Orthogonal Gradient Descent (OGD) case of Theorem 1. The key difference in the proof is that the OGD optimisation objective is constrained. The constraints correspond to the orthogonality to the subspace spanned by the memorised feature maps from the previous tasks. Another key difference occurs during the regularisation, as opposed to the SGD case where the regularisation of the weights spans on the whole space, the regularisation of OGD only applies to the "learnable" space. This property follows from the orthogonality constraint, which enforces that the space spanned by the previous tasks is unchanged.

**Proof**

We prove the Theorem 1 by induction. Our induction hypothesis $\mathcal{H}_\tau$ is the following : $\mathcal{H}_\tau$ : For all $k \leq \tau$, Theorem 1 holds.

First, we prove that $\mathcal{H}_1$ holds.

The proof is straightforward. For the first task, since there were no previous tasks, OGD on this task is the same as SGD.

Therefore, it is equivalent to minimising the following objective :

$$\underset{\mathbf{w}\in\mathbb{R}^d}{\arg\min} \big\| f_0(\mathbf{X}_1)\phi(\mathbf{X}_1)^T(\mathbf{w} - \mathbf{w}_0^\star) - \mathbf{y}_1 \big\|_2^2 + \frac{\lambda}{2}\|\mathbf{w} - \mathbf{w}_0^\star\|^2 \tag{16}$$

We replace into the objective by the residual term $\tilde{\mathbf{y}}_1 = y_1 - f_0(\mathbf{X}_1)$

$$\underset{\mathbf{w}\in\mathbb{R}^d}{\arg\min} \big\| \phi(\mathbf{X}_1)^T(\mathbf{w} - \mathbf{w}_0^\star) - \tilde{\mathbf{y}}_1 \big\|_2^2 + \frac{\lambda}{2}\|\mathbf{w} - \mathbf{w}_0\|^2 \tag{17}$$

where $\phi(\mathbf{x}) = \nabla_{\mathbf{w}_0^\star} f_0^\star(\mathbf{x})$.

The objective is quadratic and its Hessian is positive definite, therefore the minimum exists and is unique :

$$\mathbf{w}_1^\star - \mathbf{w}_0^\star = \phi(\mathbf{X}_1)(\phi(\mathbf{X}_1)^T\phi(\mathbf{X}_1) + \lambda\mathbf{I})^{-1}\tilde{\mathbf{y}}_1 \tag{18}$$

Under the NTK regime assumption :

$$f_1^\star(\mathbf{x}) = f_0^\star(\mathbf{x}) + \nabla_{\mathbf{w}} f_0^\star(\mathbf{x})^T(\mathbf{w}_1^\star - \mathbf{w}_0^\star) \tag{19}$$

Then, by replacing into $\mathbf{w}_1^\star - \mathbf{w}_0^\star$ :

$$f_1^\star(\mathbf{x}) = f_0^\star(\mathbf{x}) + \nabla_{\mathbf{w}} f_0^\star(\mathbf{x})^T\phi(\mathbf{X}_1)(\phi(\mathbf{X}_1)^T\phi(\mathbf{X}_1) + \lambda\mathbf{I})^{-1}\tilde{\mathbf{y}}_1 \tag{20}$$

$$f_1^\star(\mathbf{x}) = f_0^\star(\mathbf{x}) + \kappa_1(\mathbf{x}, \mathbf{X}_1)(\kappa_1(\mathbf{X}_1, \mathbf{X}_1) + \lambda\mathbf{I})^{-1}\tilde{\mathbf{y}}_1 \tag{21}$$

Finally :

$$f_1^\star(\mathbf{x}) - f_0^\star(\mathbf{x}) = \kappa_1(\mathbf{x}, \mathbf{X}_1)(\kappa_1(\mathbf{X}_1, \mathbf{X}_1) + \lambda\mathbf{I})^{-1}\tilde{\mathbf{y}}_1 \tag{22}$$

Which completes the proof of $\mathcal{H}_1$.

Let $\tau \in \mathbb{N}^\star$, we assume that $\mathcal{H}_\tau$ is true, then we show that $\mathcal{H}_{\tau+1}$ is true.

On the task $\mathcal{T}_{\tau+1}$, we can write the loss $\mathcal{L}^{\tau+1}$ as :

$$\mathcal{L}^{\tau+1}(\mathbf{w}) = \left\|\phi_\tau(\mathbf{X}_{\tau+1})^T(\mathbf{w} - \mathbf{w}_\tau^\star) - \tilde{\mathbf{y}}_{\tau+1}\right\|_2^2 + \frac{\lambda}{2}\|\mathbf{w} - \mathbf{w}_\tau^\star\|^2 \tag{23}$$

We recall that the optimisation problem on the task $\mathcal{T}_{\tau+1}$ is :

$$\underset{\mathbf{w} \in \mathbb{R}^d}{\arg\min} \left\|\phi_\tau(\mathbf{X}_{\tau+1})^T(\mathbf{w} - \mathbf{w}_\tau^\star) - \tilde{\mathbf{y}}_{\tau+1}\right\|_2^2 + \frac{\lambda}{2}\|\mathbf{w} - \mathbf{w}_\tau^\star\|^2 \tag{24}$$

$$\text{u.c.} \quad \mathbf{V}_{\tau+1}(\mathbf{w} - \mathbf{w}_\tau^\star) = 0 \tag{25}$$

where $\mathbf{V}_{\tau+1}$ is a projection matrix on $(\bigoplus_{k=1}^{\tau} \mathbb{E}_k)^\perp$, the Euclidean space induced by the feature maps of the tasks $\{\mathcal{T}_k, k \in [\tau]\}$.

Let $\mathbf{T}_{\tau+1} \in \mathbb{R}^{d \times (d-K_{\tau+1})}$ and $\tilde{\mathbf{w}} \in \mathbb{R}^{d-K_{\tau+1}}$ such as :

$$\mathbf{w} - \mathbf{w}_\tau^\star = \mathbf{T}_{\tau+1}\tilde{\mathbf{w}} \tag{26}$$

$$K_\tau = \dim(\bigoplus_{k=1}^{\tau} \mathbb{E}_k) \tag{27}$$

We rewrite the objective by plugging in the variables we just defined. The two objectives are equivalent :

$$\underset{\tilde{\mathbf{w}} \in \mathbb{R}^{d-K_{\tau+1}}}{\arg\min} \left\|\phi_\tau(\mathbf{X}_{\tau+1})^T\mathbf{T}_{\tau+1}\tilde{\mathbf{w}} - \tilde{\mathbf{y}}_{\tau+1}\right\|_2^2 + \frac{\lambda}{2}\|\mathbf{T}_{\tau+1}\tilde{\mathbf{w}}\|^2 \tag{28}$$

This objective is equivalent to :

$$\underset{\tilde{\mathbf{w}} \in \mathbb{R}^{d-K_{\tau+1}}}{\arg\min} \left\|\phi_\tau(\mathbf{X}_{\tau+1})^T\mathbf{T}_{\tau+1}\tilde{\mathbf{w}} - \tilde{\mathbf{y}}_{\tau+1}\right\|_2^2 + \frac{\lambda}{2}\|\tilde{\mathbf{w}}\|^2 \tag{29}$$

For clarity, we define $\mathbf{Z}_{\tau+1}\mathbb{R}^{n_{\tau+1} \times (d-K_{\tau+1})}$ as :

$$\mathbf{Z}_{\tau+1} = \phi_\tau(\mathbf{X}_{\tau+1})^T\mathbf{T}_{\tau+1} \tag{30}$$

By plugging in $\mathbf{Z}_{\tau+1}$, we rewrite the objective as :

$$\underset{\tilde{\mathbf{w}} \in \mathbb{R}^{d-K_{\tau+1}}}{\arg\min} \left\|\mathbf{Z}_{\tau+1}\tilde{\mathbf{w}} - \tilde{\mathbf{y}}_{\tau+1}\right\|_2^2 + \frac{\lambda}{2}\|\mathbf{T}_{\tau+1}\tilde{\mathbf{w}}\|^2 \tag{31}$$

The optimisation objective is quadratic, unconstrained, with a positive definite hessian. Therefore, an optimum exists and is unique :

$$\tilde{\mathbf{w}}_{\tau+1}^\star = \mathbf{Z}_{\tau+1}^T(\mathbf{Z}_{\tau+1}\mathbf{Z}_{\tau+1}^T + \lambda\mathbf{I})^{-1}\tilde{\mathbf{y}}_{\tau+1} \tag{32}$$

We recover the expression of the optimum in the original space :

$$\mathbf{w}_{\tau+1}^\star - \mathbf{w}_\tau^\star = \mathbf{T}_{\tau+1}\mathbf{Z}_{\tau+1}^T(\mathbf{Z}_{\tau+1}\mathbf{Z}_{\tau+1}^T + \lambda\mathbf{I})^{-1}\tilde{\mathbf{y}}_{\tau+1} \tag{33}$$

We define the kernel $\kappa_{\tau+1} : \mathbb{R}^d \times \mathbb{R}^d \to \mathbb{R}$ as :

$$\kappa_{\tau+1}(\mathbf{x}, \mathbf{x}') = \phi_\tau(\mathbf{x})^T\mathbf{T}_{\tau+1}\mathbf{T}_{\tau+1}^T\phi_\tau(\mathbf{x}') \quad \text{for all } \mathbf{x}, \mathbf{x}' \in \mathbb{R}^d \tag{34}$$

Now we rewrite $\mathbf{w}_{\tau+1}^\star - \mathbf{w}_\tau^\star$ :

$$\mathbf{w}_{\tau+1}^\star - \mathbf{w}_\tau^\star = \mathbf{T}_{\tau+1}\mathbf{Z}_{\tau+1}^T(\kappa_{\tau+1}(\mathbf{X}_{\tau+1}, \mathbf{X}_{\tau+1}) + \lambda\mathbf{I})^{-1}\tilde{\mathbf{y}}_{\tau+1} \tag{35}$$

Finally, we recover a closed form expression for $f_{\tau+1}^\star$ :

First, we use the induction hypothesis $\mathcal{H}_\tau$ :

$$f_{\tau+1}^\star(\mathbf{x}) = f_\tau^\star(\mathbf{x}) + \langle \nabla_\mathbf{w} f_\tau^\star(\mathbf{x}), \mathbf{w}_{\tau+1}^\star - \mathbf{w}_\tau^\star \rangle \tag{36}$$

$$= f_\tau^\star(\mathbf{x}) + \phi_\tau(\mathbf{x})\mathbf{T}_{\tau+1}\mathbf{Z}_{\tau+1}^T(\kappa_{\tau+1}(\mathbf{X}_{\tau+1}, \mathbf{X}_{\tau+1}) + \lambda\mathbf{I})^{-1}\tilde{\mathbf{y}}_{\tau+1} \tag{37}$$

$$= f_\tau^\star(\mathbf{x}) + \kappa_{\tau+1}(\mathbf{x}, \mathbf{X}_{\tau+1})(\kappa_{\tau+1}(\mathbf{X}_{\tau+1}, \mathbf{X}_{\tau+1}) + \lambda\mathbf{I})^{-1}\tilde{\mathbf{y}}_{\tau+1} \tag{38}$$

At this stage, we have proven $\mathcal{H}_{t+1}$.

We conclude the proof of Thm. 1.

$$\tag{39}$$

## C.2  PROOF OF THE COROLLARY 1

**Stochastic Gradient Descent**  The proof follows immediately from Thm. 1.

**Proof**

In the proof of Theorem 1 (App. C.1), for the SGD case, we showed that :

$$\mathbf{w}^\star_{\tau+1} - \mathbf{w}^\star_\tau = \phi(\mathbf{X}_{\tau+1})(\phi(\mathbf{X}_{\tau+1})^T\phi(\mathbf{X}_{\tau+1}) + \lambda\mathbf{I})^{-1}\tilde{\mathbf{y}}_{\tau+1} \tag{40}$$

This result implies that :

$$\left\|\mathbf{w}^\star_{\tau+1} - \mathbf{w}^\star_\tau\right\|^2 = \tilde{\mathbf{y}}^T_{\tau+1}(\phi(\mathbf{X}_{\tau+1})^T\phi(\mathbf{X}_{\tau+1}) + \lambda\mathbf{I})\phi(\mathbf{X}_{\tau+1})^T\phi(\mathbf{X}_{\tau+1})(\phi(\mathbf{X}_{\tau+1})^T\phi(\mathbf{X}_{\tau+1}) + \lambda\mathbf{I})^{-1}\tilde{\mathbf{y}}_{\tau+1} \tag{41}$$

$$= \tilde{\mathbf{y}}^T_{\tau+1}(\kappa(\mathbf{X}_{\tau+1}, \mathbf{X}_{\tau+1}) + \lambda\mathbf{I})^{-1}\kappa(\mathbf{X}_{\tau+1}, \mathbf{X}_{\tau+1})(\kappa(\mathbf{X}_{\tau+1}, \mathbf{X}_{\tau+1}) + \lambda\mathbf{I})^{-1}\tilde{\mathbf{y}}_{\tau+1} \tag{42}$$

$$\tag{43}$$

**Orthogonal Gradient Descent**  The proof is exactly the same as the proof above, the difference lies in the kernel definition, which is implicit.

**Proof**

In the proof of Theorem 1 (App. C.1), for the OGD case, we showed that :

$$\mathbf{w}^\star_{\tau+1} - \mathbf{w}^\star_\tau = \mathbf{T}_{\tau+1}\mathbf{Z}^T_{\tau+1}(\kappa_{\tau+1}(\mathbf{X}_{\tau+1}, \mathbf{X}_{\tau+1}) + \lambda\mathbf{I})^{-1}\tilde{\mathbf{y}}_{\tau+1} \tag{44}$$

This result implies that :

$$\left\|\mathbf{w}^\star_{\tau+1} - \mathbf{w}^\star_\tau\right\|^2 = \tilde{\mathbf{y}}^T_{\tau+1}(\kappa_{\tau+1}(\mathbf{X}_{\tau+1}, \mathbf{X}_{\tau+1}) + \lambda\mathbf{I})^{-1}\mathbf{Z}_{\tau+1}\mathbf{T}^T_{\tau+1}\mathbf{T}_{\tau+1}\mathbf{Z}^T_{\tau+1}(\kappa_{\tau+1}(\mathbf{X}_{\tau+1}, \mathbf{X}_{\tau+1}) + \lambda\mathbf{I})^{-1}\tilde{\mathbf{y}}_{\tau+1} \tag{45}$$

$$= \tilde{\mathbf{y}}^T_{\tau+1}(\kappa_{\tau+1}(\mathbf{X}_{\tau+1}, \mathbf{X}_{\tau+1}) + \lambda\mathbf{I})^{-1}\kappa_{\tau+1}(\mathbf{X}_{\tau+1}, \mathbf{X}_{\tau+1})(\kappa_{\tau+1}(\mathbf{X}_{\tau+1}, \mathbf{X}_{\tau+1}) + \lambda\mathbf{I})^{-1}\tilde{\mathbf{y}}_{\tau+1} \tag{46}$$

$$\tag{47}$$

# D   MISSING PROOFS OF SECTION 5 - GENERALISATION

## D.1   PROOF OF THEOREM 2

The intuition behind the proof is : since the gradient updates were performed orthogonally to the feature maps of the training data of the source task, the parameters in this space are unchanged, while the remaining space, which was changed, is orthogonal to these features maps, therefore, the inference is the same and the training error remains the same as at the end of training on the source task.

**Proof**

In the proof of Theorem 1, App. C.1, we showed that, for $\mathcal{T}_\tau$ a fixed task:

$$f_{\tau+1}^\star(\mathbf{x}) = f_\tau^\star(\mathbf{x}) + \langle \nabla_\mathbf{w} f_\tau^\star(\mathbf{x}), \mathbf{w}_{\tau+1}^\star - \mathbf{w}_\tau^\star \rangle. \tag{48}$$

We rewrite the recursive relation into a sum:

$$f_{\tau+1}^\star(\mathbf{x}) - f_0^\star(\mathbf{x}) = \sum_{k=1}^\tau \langle \nabla_\mathbf{w} f_k^\star(\mathbf{x}), \mathbf{w}_{k+1}^\star - \mathbf{w}_k^\star \rangle. \tag{49}$$

We observe that, for all $k \in [T]$:

$$\mathbf{w}_{k+1}^\star - \mathbf{w}_k^\star \in \mathbb{E}_{k'}. \tag{50}$$

On the other hand, for OGD+, given a sample $\mathbf{x}$ from $\mathcal{D}_\tau$, for all $k' \in [\tau+1, T]$ :

$$\nabla_\mathbf{w} f_k^\star(\mathbf{x}) \in \mathbb{E}_{k'} \tag{51}$$

Therefore, for all $k' \in [k+1, \tau]$ :

$$\langle \nabla_\mathbf{w} f_{k'}^\star(\mathbf{x}), \mathbf{w}_{k'+1}^\star - \mathbf{w}_{k'}^\star \rangle = 0 \tag{52}$$

Therefore :

$$f_\tau^\star(\mathbf{x}) = f_k^\star(\mathbf{x}) \tag{53}$$

We conclude.

$$\tag{54}$$

## D.2   PROOF OF THEOREM 3

### D.2.1   REMINDERS

**Reminder on RKHS norm**

Let $\kappa$ a kernel, and $\mathcal{H}$ the reproducing kernel Hilbert space (RKHS) corresponding to the kernel $\kappa$.

Recall that the RKHS norm of a function $f(\mathbf{x}) = \alpha^T \kappa(\mathbf{x}, \mathbf{X})$ is :

$$\|f\|_\mathcal{H} = \sqrt{\alpha^T \kappa(\mathbf{X}, \mathbf{X}) \alpha} \tag{55}$$

$$\tag{56}$$

**Reminder on Generalization and Rademacher Complexity**   Consider a loss function $l : \mathbb{R} \times \mathbb{R} \to \mathbb{R}$. The population loss over the distribution $\mathcal{D}$, and the empirical loss over $n$ samples $D = \{(\mathbf{x}_i, y_i), i \in [n]\}$ from the same distribution $\mathcal{D}$ are defined as :

$$L_D(f) = \mathbb{E}_{(\mathbf{x},y) \sim \mathcal{D}}[l(f(\mathbf{x}), y)] \tag{57}$$

$$L_S(f) = \frac{1}{n} \sum_{i=1}^n l(f(\mathbf{x}_i), y_i) \tag{58}$$

**Theorem 4** *Suppose the loss function is bounded in $[0, c]$ and is $\rho-$Lipschitz in the first argument. Then, with probability at least $1 - \delta$ over sample $S$ of size $n$ :*

$$\sup_{f \in \mathcal{F}} \{L_D(f) - L_S(f)\} \le 2\rho \hat{\mathcal{R}}(\mathcal{F}) + 3c\sqrt{\frac{\log(2/\delta)}{2n}} \tag{59}$$

### D.2.2 Bounding $\left\|\tilde{f}_\tau^\star\right\|_{\mathcal{H}_\tau}$ :

**Lemma 2** *Let $\mathcal{H}_\tau$ the Hilbert space associated to the kernel $\kappa_\tau$ .*

*We recall that :*

$$\tilde{f}_\tau^\star(\mathbf{x}) = \kappa_\tau(\mathbf{x}, \mathbf{X}_\tau)^T \boldsymbol{\alpha}_\tau \tag{60}$$

$$\boldsymbol{\alpha}_\tau = (\kappa_\tau(\mathbf{X}_\tau, \mathbf{X}_\tau) + \lambda\mathbf{I})^{-1}\tilde{\mathbf{y}}_\tau \tag{61}$$

*Then :*

$$\left\|\tilde{f}_\tau^\star\right\|_{\mathcal{H}_\tau}^2 \le \tilde{\mathbf{y}}_\tau^T (\kappa_\tau(\mathbf{X}_\tau, \mathbf{X}_\tau) + \lambda\mathbf{I})^{-1}\tilde{\mathbf{y}}_\tau \tag{62}$$

$$\tag{63}$$

**Proof**

We start from the definition of the RKHS norm of $\tilde{f}_\tau^\star$ :

$$\left\|\tilde{f}_\tau^\star\right\|_{\mathcal{H}_\tau}^2 = \boldsymbol{\alpha}_\tau^T \kappa_\tau(\mathbf{X}_\tau, \mathbf{X}_\tau)\boldsymbol{\alpha}_\tau \tag{64}$$

$$= \tilde{\mathbf{y}}_\tau^T(\kappa_\tau(\mathbf{X}_\tau, \mathbf{X}_\tau) + \lambda\mathbf{I})^{-1}\kappa_\tau(\mathbf{X}_\tau, \mathbf{X}_\tau)(\kappa_\tau(\mathbf{X}_\tau, \mathbf{X}_\tau) + \lambda\mathbf{I})^{-1}\tilde{\mathbf{y}}_\tau \tag{65}$$

Since $(\kappa_\tau(\mathbf{X}_\tau, \mathbf{X}_\tau) + \lambda\mathbf{I})^{-1} \le (\kappa_\tau(\mathbf{X}_\tau, \mathbf{X}_\tau))^{-1}$

$$\le \tilde{\mathbf{y}}_\tau^T(\kappa_\tau(\mathbf{X}_\tau, \mathbf{X}_\tau) + \lambda\mathbf{I})^{-1}\kappa_\tau(\mathbf{X}_\tau, \mathbf{X}_\tau)\kappa_\tau(\mathbf{X}_\tau, \mathbf{X}_\tau)^{-1}\tilde{\mathbf{y}}_\tau \tag{66}$$

$$\le \tilde{\mathbf{y}}_\tau^T(\kappa_\tau(\mathbf{X}_\tau, \mathbf{X}_\tau) + \lambda\mathbf{I})^{-1}\tilde{\mathbf{y}}_\tau \tag{67}$$

### D.2.3 Bounding the Rademacher Complexity

The goal of this section is to upper bound the Rademacher Complexity. First, we derive a general upper bound for the Rademacher Complexity of a linear combination of kernels in Lemma 3 . Then, we apply this bound to the linear combination of kernels obtained through Thm. 1, which describes the model through the Continual Learning.

**A general bound for the Rademacher Complexity**

**Lemma 3 (Rademacher Complexity of a linear combination of kernels)** *Let $\kappa_t : \mathcal{X} \times \mathcal{X} \to \mathbb{R}, t \in [T]$ kernels such that :*

$$\sup_{\mathbf{x} \in \mathcal{X}} \|\kappa_t(\mathbf{x}, \mathbf{x})\| < \infty$$

*To every kernel $\kappa_t$, we associate a feature map $\phi_t : \mathcal{X} \to \mathcal{H}_t$, where $\mathcal{H}_t$ is a Hilbert space with inner product $\langle\cdot,\cdot\rangle_{\mathcal{H}_t}$, and for all $\mathbf{x}, \mathbf{x}' \in \mathcal{X}$, $\kappa_t(\mathbf{x}, \mathbf{x}') = \langle\phi_t(\mathbf{x}), \phi_t(\mathbf{x}')\rangle_{\mathcal{H}_t}$*

*For all $T \in \mathbb{N}^\star$, given a sequence $\{B_\tau, \tau \in [T]\}$ we define $\mathcal{F}_T$ as follows :*

$$\mathcal{F}_T = \{\mathbf{x} \to \sum_{t=1}^T f_t(\mathbf{x}), \quad f_t(\mathbf{x}) = \alpha_t^T \kappa_t(\mathbf{x}, \mathbf{X}_t) \quad \forall t \in [T], \|f_t\|_{\mathcal{H}_t} \le B_t\} \tag{68}$$

*Let $X_1, \cdot, X_n$ be random elements of $\mathcal{X}$. Then for the class $\mathcal{F}$, we have :*

$$\hat{\mathcal{R}}(\mathcal{F}) \le \sum_{t=1}^T \frac{2B_t}{n_t}(Tr(\kappa_t(\mathbf{X}_t, \mathbf{X}_t)))^{1/2}$$

**Proof**

Let $f \in \mathcal{F}$, and let $\mathbf{x} \in \mathcal{X}$ :

$$f(\mathbf{x}) = \sum_{\tau=1}^{T} \sum_{i=1}^{n_\tau} \alpha_i^\tau \kappa_\tau(\mathbf{x}, \mathbf{x}_i^\tau) \tag{69}$$

For all $\tau \in [T]$, we associate a feature map $\phi_\tau : \mathbf{X} \to \mathcal{H}_\tau$

$$\forall \mathbf{x}, \mathbf{x}' \in \mathcal{X} \quad \kappa_\tau(\mathbf{x}, \mathbf{x}') = \langle \phi_\tau(\mathbf{x}), \phi_\tau(\mathbf{x}') \rangle_{\mathcal{H}_\tau} \tag{70}$$

Therefore :

$$f(\mathbf{x}) = \sum_{t=1}^{T} \sum_{i=1}^{n_\tau} \alpha_i^\tau \langle \phi_\tau(\mathbf{x}_i^\tau), \phi_\tau(\mathbf{x}) \rangle_{\mathcal{H}_\tau} \tag{71}$$

$$= \sum_{\tau=1}^{T} \langle \sum_{i=1}^{n_\tau} \alpha_i^\tau \phi_\tau(\mathbf{x}_i^\tau), \phi_\tau(\mathbf{x}) \rangle_{\mathcal{H}_\tau} \tag{72}$$

On the other hand, the following holds $\forall t \in [T]$ :

$$\left\| \sum_{i=1}^{n_\tau} \alpha_i^\tau \phi_\tau(\mathbf{x}_i^\tau) \right\|_{\mathcal{H}_\tau}^2 = \sum_{i,j} \alpha_i^\tau \alpha_j^\tau \kappa_\tau(\mathbf{x}_i^\tau, \mathbf{x}_j^\tau) \leq B_\tau^2 \tag{73}$$

Therefore :

$$\mathcal{F}_T \subset \{\mathbf{x} \to \sum_{t=1}^{T} \langle \mathbf{w}_\tau, \phi_\tau(\mathbf{x}) \rangle_{\mathcal{H}_\tau}, \|\mathbf{w}_\tau\|_2 \leq B_\tau \quad \forall t \in [T]\} := \tilde{\mathcal{F}}_T \tag{74}$$

Now, we derive an upper bound of the Rademacher complexity of $\mathcal{F}_T$ :

$$\hat{\mathcal{R}}(\mathcal{F}_T) \leq \hat{\mathcal{R}}(\tilde{\mathcal{F}}_T) \tag{75}$$

$$= \mathbb{E}[\sup_{\|\mathbf{W}_\tau\|_2 \leq B_\tau, t \in [T]} \sum_{t=1}^{T} \langle \mathbf{w}_\tau, \frac{2}{n_\tau} \sum_{i=1}^{n_\tau} \epsilon_i \phi_\tau(\mathbf{x}_i^\tau) \rangle_{\mathcal{H}_\tau} | (\mathbf{X}_\tau)] \tag{76}$$

$$= \sum_{t=1}^{T} \mathbb{E}[\sup_{\|\mathbf{W}_\tau\|_2 \leq B_\tau} \langle \mathbf{w}_\tau, \frac{2}{n_\tau} \sum_{i=1}^{n_\tau} \epsilon_i \phi_\tau(\mathbf{x}_i^\tau) \rangle_{\mathcal{H}_\tau} | (\mathbf{X}_\tau)] \tag{77}$$

Now we apply on each function $f_\tau$ the upper bound from Lemma 22 by Bartlett & Mendelson 2003

$$\hat{\mathcal{R}}(\mathcal{F}_T) \leq \sum_{t=1}^{T} \frac{2 B_\tau}{n_\tau} (Tr(\kappa_\tau(\mathbf{X}_\tau, \mathbf{X}_\tau)))^{1/2} \tag{78}$$

$$\tag{79}$$

### Bounding the Rademacher Complexity for Continual Learning

**Lemma 4** *Keeping the same notations and setting as Theorem 3, the Rademacher Complexity can be bounded as follows:*

$$\hat{\mathcal{R}}(\mathcal{F}_T) \leq \sum_{\tau=1}^{T} \mathcal{O}\left( \sqrt{\frac{\tilde{\mathbf{y}}_\tau^T (\kappa_\tau(\mathbf{X}_\tau, \mathbf{X}_\tau) + \lambda \mathbf{I})^{-1} \tilde{\mathbf{y}}_\tau}{n_\tau}} \right), \tag{80}$$

*where $\mathcal{F}_T$ is the function class spanned by the model up to the task $\mathcal{T}_\tau$.*

$$\tag{81}$$

### Proof

For all $T \in \mathbb{N}^\star$, given a sequence $\{B_\tau, \tau \in [T]\}$ we define $\mathcal{F}_T$ as follows :

$$\mathcal{F}_T = \{\mathbf{x} \to \sum_{\tau=1}^{T} f_\tau(\mathbf{x}), \quad f_\tau(\mathbf{x}) = \alpha_\tau^T \kappa_\tau(\mathbf{x}, \mathbf{X}_\tau) \quad \forall \tau \in [T], \|f_\tau\|_{\mathcal{H}_\tau} \leq B_\tau\} \tag{82}$$

where we set $B_\tau$ as :

$$B_\tau = \sqrt{(\mathbf{y}_\tau - \mathbf{y}_{\tau-1\to\tau})^T(\kappa_\tau(\mathbf{X}_\tau, \mathbf{X}_\tau) + \lambda\mathbf{I})^{-1}(\mathbf{y}_\tau - \mathbf{y}_{\tau-1\to\tau})} \tag{83}$$

Following Lemma 2 for all $\tau \in [T]$ :

$$f_\tau^\star \in \mathcal{F}_T \tag{84}$$

which means that the function class $\mathcal{F}_t$ contains the Continual Learning model up to task $\mathcal{T}_\tau$.

We apply Lemma 3 in order to upper bound the Rademacher complexity :

$$\hat{\mathcal{R}}(\mathcal{F}_T) \leq \sum_{\tau=1}^{T} \frac{2B_\tau}{n_\tau}(Tr(\kappa_\tau(\mathbf{X}_\tau, \mathbf{X}_\tau)))^{1/2} \tag{85}$$

We made the assumption that for all $\tau \in [T]$ $\mathrm{tr}(\kappa_\tau(\mathbf{X}_\tau, \mathbf{X}_\tau)) = \mathcal{O}(n_\tau)$ :

$$\hat{\mathcal{R}}(\mathcal{F}_T) \leq \sum_{\tau=1}^{T} \frac{2B_\tau}{n_\tau}\mathcal{O}(\sqrt{n_\tau}) \tag{86}$$

$$\leq \sum_{\tau=1}^{T} \mathcal{O}(\frac{B_\tau}{\sqrt{n_\tau}}) \tag{87}$$

$$\leq \sum_{\tau=1}^{T} \mathcal{O}\left(\sqrt{\frac{(\mathbf{y}_\tau - \mathbf{y}_{\tau-1\to\tau})^T(\kappa_\tau(\mathbf{X}_\tau, \mathbf{X}_\tau) + \lambda\mathbf{I})^{-1}(\mathbf{y}_\tau - \mathbf{y}_{\tau-1\to\tau})}{n_\tau}}\right) \tag{88}$$

$$\tag{89}$$

### D.2.4 BOUNDING THE EMPIRICAL LOSS FOR OGD

**Lemma 5** *Given a current task $\mathcal{T}_T$, the empirical losses on the data from all previous tasks ($\mathcal{T}_\tau, \tau \leq T$) can be bounded as follows :*

*Let $T \in \mathbb{N}$ fixed. Then, for all $\tau \in [T]$*

$$\mathcal{L}_{S_\tau}(f_T^\star) \leq \frac{\lambda^2}{n_\tau}\tilde{\mathbf{y}}_\tau(\kappa_\tau(\mathbf{X}_\tau, \mathbf{X}_\tau) + \lambda\mathbf{I})^{-1}\tilde{\mathbf{y}}_\tau \tag{90}$$

$$\tag{91}$$

**Case 1 -Task $\mathcal{T}_T$    Proof**

We start from the definition of the empirical loss :

$$\mathcal{L}_{S_T}(f_T^\star) = \frac{1}{n_T}\sum_{i=1}^{n_T}(f_T^\star(\mathbf{x}_{T,i}) - y_{T,i})^2 \tag{92}$$

$$= \frac{1}{n_T}\|f_T^\star(\mathbf{X}_T) - \mathbf{y}_T\|_2^2 \tag{93}$$

We replace into $f_T^\star$ by its expression from Theorem 1 :

$$= \frac{1}{n_T}\left\|\tilde{f}_T^\star(\mathbf{X}_T) + f_{T-1}^\star(\mathbf{X}_T) - \mathbf{y}_T\right\|_2^2 \tag{94}$$

$$= \frac{1}{n_T}\left\|f_{T-1}^\star(\mathbf{X}_T) - \tilde{\mathbf{y}}_T\right\|_2^2 \tag{95}$$

$$= \frac{1}{n_T}\left\|(k_T(\mathbf{X}_T, \mathbf{X}_T)^T(\kappa_T(\mathbf{X}_T, \mathbf{X}_T) + \lambda\mathbf{I})^{-1}\tilde{\mathbf{y}}_T - \tilde{\mathbf{y}}_T\right\|_2^2 \tag{96}$$

$$= \frac{1}{n_T}\left\|(k_T(\mathbf{X}_T, \mathbf{X}_T) + \lambda\mathbf{I} - \lambda\mathbf{I})(\kappa_T(\mathbf{X}_T, \mathbf{X}_T) + \lambda\mathbf{I})^{-1}\tilde{\mathbf{y}}_T - \tilde{\mathbf{y}}_T\right\|_2^2 \tag{97}$$

$$= \frac{1}{n_T}\left\|\tilde{\mathbf{y}}_T - \lambda(\kappa_T(\mathbf{X}_T, \mathbf{X}_T) + \lambda\mathbf{I})^{-1}\tilde{\mathbf{y}}_T - \tilde{\mathbf{y}}_T\right\|_2^2 \tag{98}$$

$$= \frac{\lambda^2}{n_T}\left\|(\kappa_T(\mathbf{X}_T, \mathbf{X}_T) + \lambda\mathbf{I})^{-1}\tilde{\mathbf{y}}_T\right\|_2^2 \tag{99}$$

Now, we apply Lemma 2 in order to upper bound the right hand side norm :

$$\mathcal{L}_{S_T}(f_T^\star) \leq \frac{\lambda^2}{n_T}\tilde{\mathbf{y}}_T(\kappa_T(\mathbf{X}_T, \mathbf{X}_T) + \lambda\mathbf{I})^{-1}\tilde{\mathbf{y}}_T \tag{100}$$

$$\tag{101}$$

**Case 2 - Tasks** $\{\mathcal{T}_\tau, \tau \in [1, T-1]\}$ **:** The proof is very similar to Case 1, we apply Theorem 2, which is the key property of OGD in terms of no forgetting.

**Proof**

We start from the definition of the empirical loss :

$$\mathcal{L}_{S_\tau}(f_T^\star) = \frac{1}{n_\tau}\sum_{i=1}^{n_\tau}(f_T^\star(\mathbf{x}_{\tau,i}) - y_{\tau,i})^2 \tag{102}$$

$$= \frac{1}{n_\tau}\|f_T^\star(\mathbf{X}_\tau) - \mathbf{y}_\tau\|_2^2 \tag{103}$$

Now, applying Theorem 2 which implies that $f_\tau^\star(\mathbf{X}_\tau) = f_T^\star(\mathbf{X}_\tau)$ :

$$\mathcal{L}_{S_\tau}(f_T^\star) = \frac{1}{n_\tau}\|f_\tau^\star(\mathbf{X}_\tau) - \mathbf{y}_\tau\|_2^2 \tag{104}$$

Then with a similar analysis as Case 1, we get :

$$\mathcal{L}_{S_\tau}(f_T^\star) \leq \frac{\lambda^2}{n_\tau}\tilde{\mathbf{y}}_\tau(k_T(\mathbf{X}_\tau, \mathbf{X}_\tau) + \lambda\mathbf{I})^{-1}\tilde{\mathbf{y}}_\tau \tag{105}$$

$$\tag{106}$$

### D.2.5 BOUNDING THE EMPIRICAL LOSS FOR SGD

As opposed to the analysis for OGD, forgetting can occur for OGD and Theorem 2 is no longer valid. Similarly to the OGD empirical loss analysis, we study the case on the data from the last task and the case on the data from all the other tasks.

**Lemma 6** *The empirical losses on the source and target tasks can be bounded as follows :*

*Let $T \in \mathbb{N}$ fixed. Then, for all $\tau \in [T]$*

$$\mathcal{L}_{S_T}(f_T^\star) \leq \frac{\lambda^2}{n_T}\tilde{\mathbf{y}}_T(k_T(\mathbf{X}_T, \mathbf{X}_T) + \lambda\mathbf{I})^{-1}\tilde{\mathbf{y}}_T \tag{107}$$

*For all $\tau \in [T-1]$*

$$\mathcal{L}_{S_\tau}(f_T^\star) \leq \frac{1}{n_\tau}(\lambda^2\tilde{\mathbf{y}}_\tau^T(k_\tau(\mathbf{X}_\tau, \mathbf{X}_\tau))^{-1}\tilde{\mathbf{y}}_\tau \tag{108}$$

$$+ \sum_{k=\tau+1}^{T}\tilde{\mathbf{y}}_k^T(\kappa_k(\mathbf{X}_k, \mathbf{X}_k) + \lambda\mathbf{I})^{-1}\kappa_k(\mathbf{X}_\tau, \mathbf{X}_k)\kappa_k(\mathbf{X}_\tau, \mathbf{X}_k)^T(\kappa_k(\mathbf{X}_k, \mathbf{X}_k) + \lambda\mathbf{I})^{-1}\tilde{\mathbf{y}}_k) \tag{109}$$

**Case 1 -Task** $\mathcal{T}_T$ For this case, the analysis is the same as for OGD, as no forgetting occurs with respect to the data the model is being trained on.

**Case 2 - Tasks** $\{\mathcal{T}_\tau, \tau \in [1, T-1]\}$ **:** Since SGD is not guaranteed to be robust to Catastrophic Forgetting, this case comprises additional residual terms that correspond to forgetting.

**Proof**

Let $\tau \in [T-1]$.

From Corr. 1, we recall that :

$$f_T^\star(\mathbf{x}) = f_\tau^\star(\mathbf{x}) + \sum_{k=\tau+1}^T \tilde{f}_k^\star(\mathbf{x}) \tag{110}$$

Then :

$$\|f_T^\star(\mathbf{X}_\tau) - \mathbf{y}_\tau\|_2^2 = \left\| f_\tau^\star(\mathbf{X}_\tau) + \sum_{k=\tau+1}^T \tilde{f}_k^\star(\mathbf{X}_\tau) - \mathbf{y}_\tau \right\|_2^2 \tag{111}$$

$$\leq \|f_\tau^\star(\mathbf{X}_\tau) - \mathbf{y}_\tau\|_2^2 + \left\| \sum_{k=\tau+1}^T \tilde{f}_k^\star(\mathbf{X}_\tau) \right\|_2^2 \tag{112}$$

$$\leq \underbrace{\|f_\tau^\star(\mathbf{X}_\tau) - \mathbf{y}_\tau\|_2^2}_{\text{(A)}} + \underbrace{\sum_{k=\tau+1}^T \left\| \tilde{f}_k^\star(\mathbf{X}_\tau) \right\|_2^2}_{\text{(B)}} \tag{113}$$

We can upper bound (A) similarly to the previous paragraphs :

$$\|f_\tau^\star(\mathbf{X}_\tau) - \mathbf{y}_\tau\|_2^2 \leq \lambda^2 \tilde{\mathbf{y}}_\tau^T (k_\tau(\mathbf{X}_\tau, \mathbf{X}_\tau) + \lambda \mathbf{I})^{-1} \tilde{\mathbf{y}}_\tau \tag{114}$$

Now, we upper bound (B). Let $k \in [\tau+1, T]$ :

$$\left\| \tilde{f}_k^\star(\mathbf{X}_\tau) \right\|_2^2 = \tilde{\mathbf{y}}_k^T (\kappa_k(\mathbf{X}_k, \mathbf{X}_k) + \lambda \mathbf{I})^{-1} \underbrace{\kappa_k(\mathbf{X}_\tau, \mathbf{X}_k) \kappa_k(\mathbf{X}_\tau, \mathbf{X}_k)^T}_{\text{Similarity between the tasks } \mathcal{T}_\tau \text{ and } \mathcal{T}_k} (\kappa_k(\mathbf{X}_k, \mathbf{X}_k) + \lambda \mathbf{I})^{-1} \tilde{\mathbf{y}}_k \tag{115}$$

We conclude by plugging back the upper bounds of (A) and (B)

$$\|f_T^\star(\mathbf{X}_\tau) - \mathbf{y}_\tau\|_2^2 \leq \lambda \tilde{\mathbf{y}}_\tau^T (k_\tau(\mathbf{X}_\tau, \mathbf{X}_\tau))^{-1} \tilde{\mathbf{y}}_\tau \tag{116}$$

$$+ \sum_{k=\tau+1}^T \tilde{\mathbf{y}}_k^T (\kappa_k(\mathbf{X}_k, \mathbf{X}_k) + \lambda \mathbf{I})^{-1} \kappa_k(\mathbf{X}_\tau, \mathbf{X}_k) \kappa_k(\mathbf{X}_\tau, \mathbf{X}_k)^T (\kappa_k(\mathbf{X}_k, \mathbf{X}_k) + \lambda \mathbf{I})^{-1} \tilde{\mathbf{y}}_k \tag{117}$$

Therefore :

$$\mathcal{L}_{S_\tau}(f_T^\star) \leq \frac{1}{n_\tau} (\lambda^2 \tilde{\mathbf{y}}_\tau^T (k_\tau(\mathbf{X}_\tau, \mathbf{X}_\tau))^{-1} \tilde{\mathbf{y}}_\tau \tag{118}$$

$$+ \sum_{k=\tau+1}^T \tilde{\mathbf{y}}_k^T (\kappa_k(\mathbf{X}_k, \mathbf{X}_k) + \lambda \mathbf{I})^{-1} \kappa_k(\mathbf{X}_\tau, \mathbf{X}_k) \kappa_k(\mathbf{X}_\tau, \mathbf{X}_k)^T (\kappa_k(\mathbf{X}_k, \mathbf{X}_k) + \lambda \mathbf{I})^{-1} \tilde{\mathbf{y}}_k) \tag{119}$$

### D.2.6 Proof of the Generalisation Theorem (Thm. 3)

Now, we prove the Theorem 3 by applying the lemmas we developed above.

**Proof**

With probability $1 - \delta$ we have :

$$\sup_{f \in \mathcal{F}_T} \{L_D(f) - L_S(f)\} \leq 2\rho\hat{\mathcal{R}}(\mathcal{F}_T) + 3c\sqrt{\frac{\log(2/\delta)}{2n_T}} \tag{120}$$

$$\mathcal{L}_{D_\tau}(f_T^\star) \leq \mathcal{L}_{S_\tau}(f_T^\star) + 2\rho\hat{\mathcal{R}}(\mathcal{F}_T) + 3c\sqrt{\frac{\log(2/\delta)}{2n_T}} \tag{121}$$

$$\mathcal{L}_{D_\tau}(f_T^\star) \leq \mathcal{L}_{S_\tau}(f_T^\star) + \sum_{k=1}^{T} \mathcal{O}\left(\sqrt{\frac{\tilde{\mathbf{y}}_k^T(\kappa_k(\mathbf{X}_k, \mathbf{X}_k))^{-1}\tilde{\mathbf{y}}_k}{n_k}}\right) + 3c\sqrt{\frac{\log(2/\delta)}{2n_T}} \tag{122}$$

Then, by replacing into $\mathcal{L}_{S_\tau}(f_T^\star)$ with Lemmas 5 and 6, we get the upper bounds of the theorem for the various cases of SGD and OGD.

$$\tag{123}$$

### D.2.7 Alternative proof for the empirical loss bounds - No regularisation case

This section is complementary and aims to provide a better interpretation of the bounds in Theorem 3.

We prove that in the case where there is no regularisation, the training error converges to zero. This result illustrates better the intuition behind the $\lambda$ term that appears in the upper bound of the Theorem 3, which is under the regularisation assumption.

First, we derive the differential equation of the model's output dynamics in Lemma 7. Then, we apply this Lemma to prove the convergence to zero in Lemma 8. Finally, in Lemma 9, we show that for the past tasks, the training error equals to zero all the time.

**The training dynamics of OGD**

**Lemma 7 (Differential equation of the model's output dynamics)** *Let $T \in \mathbb{N}^\star$, the index of the task currently being trained on, with OGD.*

*For all $\tau < T$, defining $\widetilde{\mathbf{u}}_\tau$ as :*

$$\widetilde{\mathbf{u}}_\tau(t) = f_\tau^{(t)}(\mathbf{X}_\tau) - \sum_{k<\tau} \tilde{f}_k^\star(\mathbf{X}_\tau) = \phi_\tau(\mathbf{X}_\tau)^T(\mathbf{w}_\tau(t) - \mathbf{w}_{\tau-1}^\star) \tag{124}$$

*The model's output dynamics while training on the task $\mathcal{T}_\tau$ is as follows :*

$$\widetilde{\mathbf{u}}_\tau(t+1) - \widetilde{\mathbf{u}}_\tau(t) = -\eta\kappa_\tau(\mathbf{X}_\tau, \mathbf{X}_\tau)(\widetilde{\mathbf{u}}_\tau(t) - \tilde{\mathbf{y}}_\tau). \tag{125}$$

$$\tag{126}$$

**Proof**

Let $\tau \in \mathbb{N}^\star$, we derive the dynamics while training on the task $\mathcal{T}_\tau$, while training on this task.

First, we define :

$$\widetilde{\tilde{\mathbf{y}}}_\tau = \tilde{\mathbf{y}}_\tau + \phi_\tau(\mathbf{X}_\tau)^T\mathbf{w}_{\tau-1}^\star \tag{127}$$

We also define the projection matrix $\mathbf{P}_\tau \in \mathbb{R}^{p \times p}$ as :

$$\mathbf{P}_\tau = \mathbf{T}_\tau^T\mathbf{T}_\tau \tag{128}$$

The matrix $\mathbf{P}_\tau$ performs the projection from the original weight space $\mathbb{R}^p$ to the trainable weight space during training on task $\mathcal{T}_\tau$, which corresponds to $(\bigoplus_{k=1}^{\tau} \mathbb{E}_k)^\perp$.

Our starting point is the SGD update rule :

$$\mathbf{w}_\tau(t+1) = \mathbf{w}_\tau(t) - \eta \mathbf{P}_{\tau+1} \nabla_{\mathbf{W}} \mathcal{L}^\tau(\mathbf{w}_\tau(t)) \tag{129}$$

where :

$$\mathcal{L}^\tau(\mathbf{w}_\tau(t)) = \left\| \phi_\tau(\mathbf{X}_\tau)^T(\mathbf{w}_\tau(t) - \mathbf{w}_{\tau-1}^\star) - \tilde{\mathbf{y}}_\tau \right\|_2^2 \tag{130}$$

$$= \left\| \phi_\tau(\mathbf{X}_\tau)^T \mathbf{w}_\tau(t) - (\tilde{\mathbf{y}}_\tau + \phi_\tau(\mathbf{X}_\tau)^T \mathbf{w}_{\tau-1}^\star) \right\|_2^2 \tag{131}$$

$$= \left\| \phi_\tau(\mathbf{X}_\tau)^T \mathbf{w}_\tau(t) - \tilde{\tilde{\mathbf{y}}}_\tau \right\|_2^2 \tag{132}$$

Therefore, we get the following expression for the gradient of the loss :

$$\nabla_{\mathbf{W}} \mathcal{L}^\tau(\mathbf{w}_\tau(t)) = \phi_\tau(\mathbf{X}_\tau)(\phi_\tau(\mathbf{X}_\tau)^T \mathbf{w}_\tau(t) - \tilde{\tilde{\mathbf{y}}}_\tau) \tag{133}$$

Then, the following holds :

$$\tag{134}$$

$$\tilde{\mathbf{u}}_\tau(t+1) - \tilde{\mathbf{u}}_\tau(t) = \phi_\tau(\mathbf{X}_\tau)^T(\mathbf{w}_\tau(t+1) - \mathbf{w}_\tau(t)) \tag{135}$$

$$= \phi_\tau(\mathbf{X}_\tau)^T(-\eta \mathbf{P}_\tau \nabla_{\mathbf{W}} \mathcal{L}(\mathbf{w}_\tau(t))) \tag{136}$$

$$= \phi_\tau(\mathbf{X}_\tau)^T(-\eta \mathbf{P}_\tau \phi_\tau(\mathbf{X}_\tau)(\phi_\tau(\mathbf{X}_\tau)^T \mathbf{w}_\tau(t) - \tilde{\tilde{\mathbf{y}}}_\tau)) \tag{137}$$

$$= (\phi_\tau(\mathbf{X}_\tau)\mathbf{T}_\tau)^T(-\eta \mathbf{T}_\tau \phi_\tau(\mathbf{X}_\tau)(\phi_\tau(\mathbf{X}_\tau)^T \mathbf{w}_\tau(t) - \tilde{\tilde{\mathbf{y}}}_\tau)) \tag{138}$$

$$= -\eta \kappa_\tau(\mathbf{X}_\tau, \mathbf{X}_\tau)(\phi_\tau(\mathbf{X}_\tau)^T \mathbf{w}_\tau(t) - \tilde{\tilde{\mathbf{y}}}_\tau) \tag{139}$$

$$= -\eta \kappa_\tau(\mathbf{X}_\tau, \mathbf{X}_\tau)(\tilde{\mathbf{u}}_\tau(t) - \tilde{\mathbf{y}}_\tau) \tag{140}$$

$$= -\eta \kappa_\tau(\mathbf{X}_\tau, \mathbf{X}_\tau)(\tilde{\mathbf{u}}_\tau(t) - \tilde{\mathbf{y}}_\tau) \tag{141}$$

It follows that :

$$\tilde{\mathbf{u}}_\tau(t+1) - \tilde{\mathbf{y}}_\tau = (\mathbf{I} - \eta \kappa_\tau(\mathbf{X}_\tau, \mathbf{X}_\tau))(\tilde{\mathbf{u}}_\tau(t) - \tilde{\mathbf{y}}_\tau) \tag{142}$$

$$\tilde{\mathbf{u}}_\tau(t) - \tilde{\mathbf{y}}_\tau = -(\mathbf{I} - \eta \kappa_\tau(\mathbf{X}_\tau, \mathbf{X}_\tau))^t(\tilde{\mathbf{u}}_\tau(0) - \tilde{\mathbf{y}}_\tau) \tag{143}$$

$$\tag{144}$$

### The training error converges to zero

**Lemma 8** *For all tasks* $\mathcal{T}_\tau$, $\tau \in [T]$, *the training error* $\mathcal{L}^\tau(\mathbf{w}_\tau(t))$ *converges to* $0$ *when* $t$ *tends to infinity* .

### Proof

We start with the definition of the training error on task T :

$$\mathcal{L}_S(f_\tau^{(t)}) = \left\| f_\tau^{(t)}(\mathbf{X}_\tau) - \mathbf{y}_\tau \right\|^2 \tag{145}$$

$$= \left\| \tilde{\mathbf{u}}_\tau(t) - \tilde{\mathbf{y}}_\tau \right\|^2 \tag{146}$$

$$= \left\| (I - \eta \kappa_\tau(\mathbf{X}_\tau, \mathbf{X}_\tau))^t(\tilde{\mathbf{u}}_\tau(0) - \tilde{\mathbf{y}}_\tau) \right\| \tag{147}$$

Under the assumption that $\kappa_\tau(\mathbf{X}_\tau, \mathbf{X}_\tau)$ is positive definite, for $\eta \leq \frac{1}{\|(\kappa_\tau(\mathbf{X}_\tau, \mathbf{X}_\tau))\|}$ the eigenvalues become less than 1, therefore :

$$\lim_{t \to \infty} \mathcal{L}_S(f_\tau^{(t)}) = 0 \tag{148}$$

### The training error on the past tasks is zero

**Lemma 9** *For all tasks* $\mathcal{T}_\tau$, $\tau \in [T]$, *the training error* $\mathcal{L}_{S_\tau}(f_T^\star) = 0$.

**Proof** This lemma follow immediately from Lemma 8 which states that the training error converges to zero for OGD on all tasks and Theorem 2 which states that the training error on the past tasks is unchanged.

# E  MISSING PROOFS OF APP A - THE IMPORTANCE OF THE NTK VARIATION FOR CONTINUAL LEARNING

## E.1  PROOF OF PROPOSITION 1 - OGD IMPLIES A-GEM-NT

We present the proof of Proposition 1, which relies mainly on Theorem 2 stating the robustness of OGD to Catastrophic Forgetting.

**Proof**

Let $\mathcal{T}_\tau$ a task fixed, we recall that in the NTK regime, the model can be expressed as :

$$f_{\mathbf{w}}(\mathbf{x}) = f_{\tau-1}^{\star}(\mathbf{x}) + \nabla f_0^{\star}(\mathbf{x})(\mathbf{w} - \mathbf{w}_{\tau-1}^{\star}) \tag{149}$$

Given a task $\mathcal{T}_k$ and its associated memory $\mathcal{M}_k$, we recall that the loss can be expressed as :

$$l(f_{\mathbf{w}}, \mathcal{M}_k) = \sum_{(\mathbf{x},y)\in\mathcal{M}_k} (f_{\tau-1}^{\star}(\mathbf{x}) + \nabla f_0^{\star}(\mathbf{x})(\mathbf{w} - \mathbf{w}_{\tau-1}^{\star}) - y)^2 \tag{150}$$

Similarly to the A-GEM paper, we define the gradients vectors $\mathbf{g}$ and $\{\mathbf{g}_k, k \in [\tau - 1]\}$ as :

$$\mathbf{g} = \nabla_{\mathbf{w}} l(f_{\mathbf{w}}, D_\tau) \tag{151}$$

$$\mathbf{g}_k = \nabla_{\mathbf{w}} l(f_{\mathbf{w}}, \mathcal{M}_k) \tag{152}$$

Now we extend the expressions :

$$\mathbf{g}_k = \nabla_{\mathbf{w}} l(f_{\mathbf{w}}, D_\tau) \tag{153}$$

$$= \nabla_{\mathbf{w}} \sum_{(\mathbf{x},y)\in\mathcal{M}_k} (f_{\tau-1}^{\star}(\mathbf{x}) + \nabla f_0^{\star}(\mathbf{x})(\mathbf{w} - \mathbf{w}_{\tau-1}^{\star}) - y)^2 \tag{154}$$

$$= \sum_{(\mathbf{x},y)\in\mathcal{M}_k} (f_{\tau-1}^{\star}(\mathbf{x}) + \nabla f_0^{\star}(\mathbf{x})(\mathbf{w} - \mathbf{w}_{\tau-1}^{\star}) - y)\nabla f_0^{\star}(\mathbf{x}) \tag{155}$$

For OGD, following Thm. 2 variant for finite memory :

$$= \sum_{(\mathbf{x},y)\in\mathcal{M}_k} (\nabla f_0^{\star}(\mathbf{x})(\mathbf{w} - \mathbf{w}_{\tau-1}^{\star}))\nabla f_0^{\star}(\mathbf{x}) \tag{156}$$

Finally, since the gradient updates are orthogonal to $\mathcal{M}_k$ :

$$\mathbf{g}_k = 0 \tag{157}$$

Therefore, for all $k \in [\tau - 1]$ :

$$\mathbf{g}_k \cdot \mathbf{g} = 0 \tag{158}$$

We conclude, OGD implies learning with GEM and A-GEM with no Positive Backward Transfer.

$$\tag{159}$$

# F EXPERIMENTS

## F.1 REPRODUCIBILITY

### F.1.1 ARCHITECTURES

Due to the memory limitations encountered while running OGD and OGD+ on the CIFAR100 and CUB200 dataset, we use smaller architectures which are different from Jung et al. (2020). For the MNIST benchmark, we keep the same architecture as Farajtabar et al. (2019).

**MNIST :** Similarly to Farajtabar et al. (2019), the neural network is a three-layer MLP with 100 hidden units in two layers, each layer uses RELU activation function. The model has 10 logits, which do not use any activation function.

**CIFAR100 :** The neural network is a multi-head LeNet Lecun et al. (1998) network with Batch Normalisation Ioffe & Szegedy (2015) and 200 hidden units for the penultimate layer.

**CUB200 :** Similarly to Jung et al. (2020), our base architecture is AlexNet (Krizhevsky et al., 2012). In order to scale for the OGD type algorithms, we changed the classifier module to a smaller 3 layer RELU neural network, with dropout (Table 5).

| Layer |
|---|
| Linear (4096, 256) |
| RELU |
| Dropout (0.5) |
| Linear (256, 128) |
| RELU |
| Linear (128, ...) |

Table 5: Classifier module of the architecture used for the CUB200 benchmark.

### F.1.2 EXPERIMENT SETUP

We run each experiment 5 times, the seeds set is the same across all experiments sets. We report the mean and standard deviation of the measurements.

### F.1.3 OGD+ IMPLEMENTATION DETAILS

**Memory :** During the OGD+ memory update step, for each task, the new associated feature maps replace the memory slots of the previous feature maps for the same task. The goal is to ensure a balance of the feature maps from all tasks in the memory.

**Multi-head models :** For the dataset streams Split MNIST and Split CIFAR-100, we consider multi-headed neural networks. We only store the feature maps with respect to the shared weights, the projection step is not performed for the heads' weights.

### F.1.4 HYPERPARAMETERS

We use the the same hyperparameters as Farajtabar et al. (2019) for the algorithms SGD, OGD and OGD+, on the MNIST benchmarks. We also keep a small learning rate, in order to preserve the locality assumption of OGD, and in order to verify the conditions of the theorems. We report the shared hyperparameters across the benchmarks and algorithms in Table 6.

For the other benchmarks and algorithms, we report the grid search space in Sec. F.1.5.

### F.1.5 HYPERPARAMETER SEARCH

We present our hyperparameter search ranges for each Continual Learning method. We selected the hyperparameter sets that maximise the average accuracy. The results and scripts of the hyperparameter search and the best hyperparameters are provided on the corresponding github repository.

| Hyperparameter | MNIST | CIFAR-100 | CUB-200 |
|---|---|---|---|
| Epochs | 5 | 50 | 75 |
| Architecture | MLP | LeNet | AlexNet (variant) |
| Hidden dimension | 100 | 200 | 256 |
| Torch seeds | | 1 to 5 | |
| Memory size per task | | 100 | |
| Activation | | RELU | |

Table 6: Hyperparameters used across experiments

**MNIST Benchmarks**

- EWC, SI and MAS : we fixed the seed to 0, then performed a grid search over the regularisation parameter in [0.1, 1, 10, 100, 1.000, 10.000]

- SGD, OGD and OGD+ : we used the same hyperparameters as Farajtabar et al. (2019).

- Stable SGD : we fixed the seed to 0 then performed a grid search over all combinations of :
    - gamma : [0.5, 0.6, 0.7, 0.8, 0.9]
    - learning rate : [0.01, 0.1, 0.25]
    - batch size : [10, 32, 64]
    - dropout [0, 0.1, 0.2, 0.3, 0.4, 0.5]

**CIFAR100 and CUB200 Benchmarks**

- EWC, SI and MAS : we fixed the seed to 1, then performed a grid search over :
    - regularisation coefficient : [0.1, 1, 10, 100, 1.000, 10.000]
    - learning rate : [0.00001, 0.001, 0.01, 0.1]
    - batch size : [32, 64, 256]

- SGD, OGD and OGD+ : For the MNIST benchmarks, we used the same hyperparameters as Farajtabar et al. (2019). While for the CIFAR100 and CUB200 benchmarks, we run the following grid search :
    - learning rate : [0.00001, 0.001, 0.01, 0.1]
    - batch size : [32, 64, 256]
    - epochs : [1, 20, 50]

- Stable SGD : we performed the following grid search :
    - gamma : [0.5, 0.6, 0.7, 0.8, 0.9]
    - learning rate : [0.001, 0.01, 0.1]
    - batch size : [10, 64]
    - dropout [0, 0.1, 0.2, 0.3, 0.4, 0.5]
    - epochs : [1, 10, 50]

### F.2 EXPERIMENTS : MEMORISATION PROPERTY OF OGD (THM. 2)

#### F.2.1 THE IMPORTANCE OF THE OVERPARAMETERIZATION ASSUMPTION

Thm. 2 states that in the NTK regime, given an infinite memory, the train error of OGD is unchanged.

**Experiments** We track the variation of the **train** accuracy of the samples in the memory through tasks after being trained on all subsequent tasks. We consider the MLP hidden layer size as a proxy for overparameterization. We run the experiments on the Permuted MNIST and Rotated MNIST benchmarks, with the OGD algorithm. We vary the hidden size from 100 to 400.

**Results** Figure 3 shows that the variation of the train accuracy of OGD+ decreases uniformly with the hidden size. This result indicates the forgetting of OGD decreases with overparameterization.

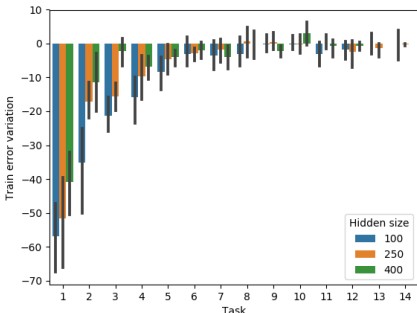

Figure 3: The variation of the train accuracy on the memorised samples from the each task, after the model was trained on all tasks in sequence (higher is better). We vary the hidden size as a proxy for overparameterization.

#### F.2.2 THE IMPORTANCE OF THE MEMORY SIZE ASSUMPTION

Thm. 2 states that in the NTK regime, given an infinite memory, the train error of OGD is unchanged.

**Experiments** We track the variation of the **train** accuracy of the samples in the memory through tasks after being trained on all subsequent tasks, as a function of the memory size per task.

**Results** Figure 4 shows that the *mean* train accuracy variation decreases uniformly with the memory size.

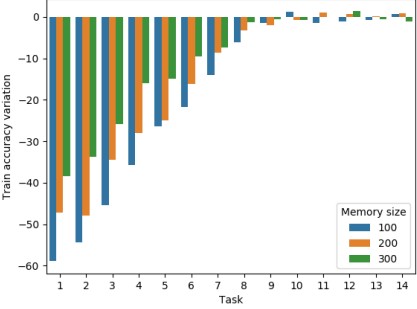

Figure 4: The variation of the train accuracy on the memorised samples from the first task, after the model was trained on all tasks in sequence (higher is better). We vary the memory size per task from 100 to 300.

### F.3 The importance of the NTK variation for Continual Learning (Sec. 6)

In this section, we present extended results on the importance of the NTK variation for Continual Learning. For completion, we present the variation of the test accuracy of the model at the end of training for all tasks. Also, we run the benchmarks on the A-GEM algorithm as a supporting materials for the discussion in App. A.

### F.3.1 Permuted MNIST

| | Accuracy ± Std. (%) | | | | | | |
|---|---|---|---|---|---|---|---|
| | Task 1 | Task 2 | Task 3 | Task 4 | Task 5 | Task 6 | Task 7 |
| A-GEM | **76.0±1.6** | **78.4±1.7** | **80.3±1.4** | **82.0±1.6** | 82.1±1.5 | **84.6±0.9** | 86.2±0.9 |
| OGD+ | 72.8±1.6 | 75.2±2.9 | 78.0±1.8 | 79.8±1.4 | 81.4±1.4 | 84.3±1.8 | 85.4±1.9 |
| OGD | 33.1±9.2 | 62.8±1.6 | 75.5±2.8 | 77.1±4.4 | **82.2±1.4** | 83.5±1.9 | **86.5±1.3** |

| | Accuracy ± Std. (%) | | | | | | |
|---|---|---|---|---|---|---|---|
| | Task 8 | Task 9 | Task 10 | Task 11 | Task 12 | Task 13 | Task 14 | Task 15 |
| A-GEM | 87.1±1.1 | **89.1±0.9** | 89.0±1.1 | 90.3±0.7 | 91.5±1.0 | 92.5±0.8 | 93.6±0.4 | **94.7±0.2** |
| OGD+ | **87.3±1.1** | 88.4±1.2 | 90.2±0.7 | 91.5±0.8 | 92.4±0.5 | **93.3±0.4** | **94.0±0.3** | 94.5±0.1 |
| OGD | 86.8±0.5 | 88.4±1.3 | **90.2±0.9** | **91.6±0.4** | **92.7±0.2** | 93.2±0.3 | 94.0±0.1 | 94.4±0.1 |

Table 7: Permuted MNIST : The test accuracy of models from the indicated task after being trained on all tasks in sequence. The best Continual Learning results are highlighted in **bold**.

### F.3.2 Rotated MNIST

| | Accuracy ± Std. (%) | | | | | | |
|---|---|---|---|---|---|---|---|
| | Task 1 | Task 2 | Task 3 | Task 4 | Task 5 | Task 6 | Task 7 |
| A-GEM | **76.1±2.3** | **78.3±2.1** | **83.6±1.1** | **86.1±0.8** | **88.1±0.5** | **89.5±0.1** | **91.0±0.3** |
| OGD+ | 65.4±1.2 | 67.8±1.1 | 75.6±1.1 | 80.1±1.5 | 83.9±1.1 | 86.3±1.2 | 88.7±0.8 |
| OGD | 41.7±1.6 | 44.7±1.4 | 53.2±1.7 | 61.4±1.4 | 68.7±1.2 | 74.8±1.2 | 81.3±0.8 |

| | Accuracy ± Std. (%) | | | | | | |
|---|---|---|---|---|---|---|---|
| | Task 8 | Task 9 | Task 10 | Task 11 | Task 12 | Task 13 | Task 14 | Task 15 |
| A-GEM | **91.6±0.2** | **93.0±0.2** | **94.0±0.2** | **95.1±0.2** | **95.9±0.0** | **96.4±0.1** | 96.3±0.1 | 96.0±0.1 |
| OGD+ | 90.3±0.7 | 92.3±0.4 | 93.9±0.2 | 94.9±0.1 | 95.9±0.1 | 96.1±0.1 | 96.1±0.1 | 95.6±0.1 |
| OGD | 86.0±0.7 | 89.7±0.5 | 92.5±0.2 | 94.4±0.1 | 95.7±0.1 | 96.1±0.2 | 96.2±0.2 | 95.8±0.2 |

Table 8: Rotated MNIST : The test accuracy of models from the indicated task after being trained on all tasks in sequence. The best Continual Learning results are highlighted in **bold**.

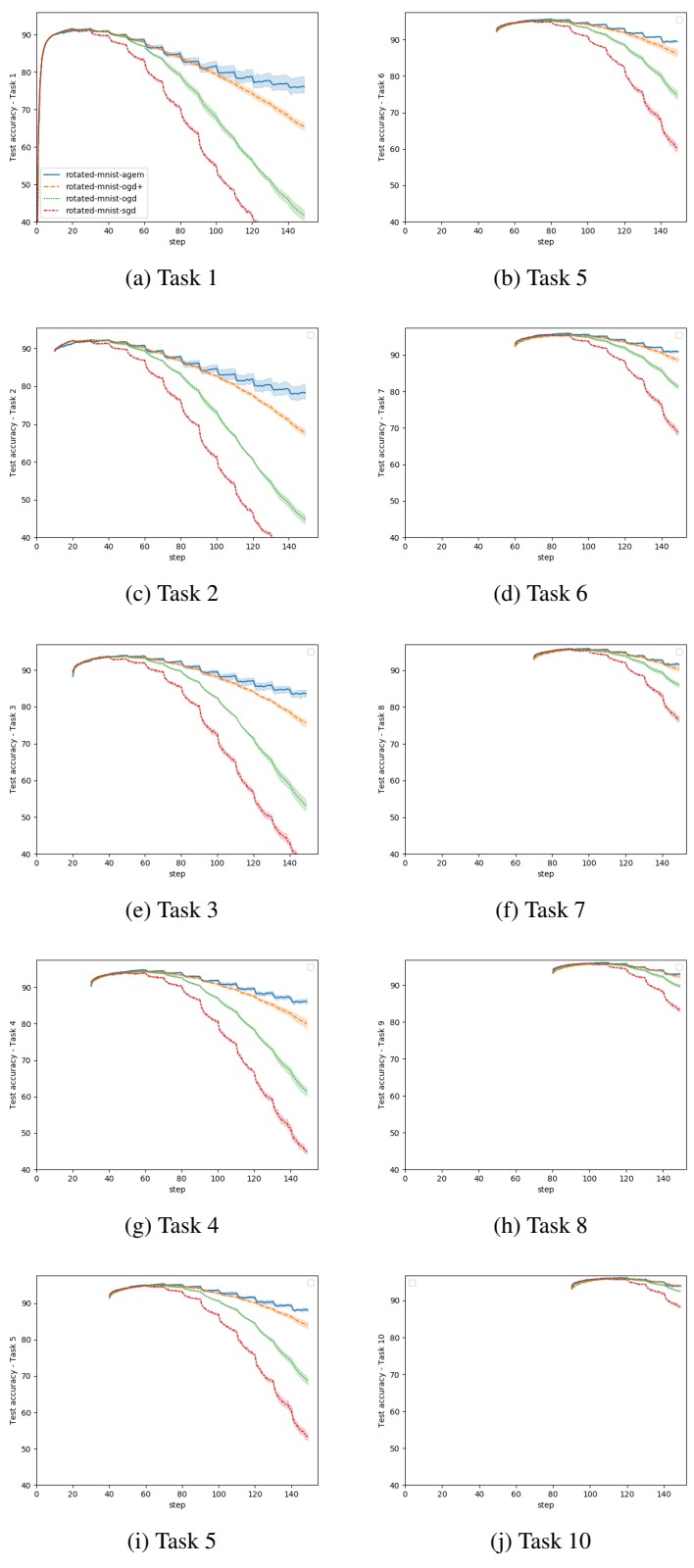

Figure 5: Test accuracy on the 10 first tasks of Rotated MNIST, for SGD, OGD, OGD+ and A-GEM. The y-axis is truncated for clarity. We report the mean and standard deviation over 5 independent runs. The test error is measured for every 250 mini-batch interval.

### F.4 COMPARISON WITH OTHER CONTINUAL LEARNING BASELINES

In order to get a broader picture on the robustness of OGD+ to Catastrophic Forgetting, we benchmark it against standard Continual Learning baselines.

#### F.4.1 EVALUATION

Denoting $a_{T,\tau}$ the accuracy of the model on the task $\mathcal{T}_\tau$ after being trained on the task $\mathcal{T}_T$, and $\bar{b}$ the vector of test accuracies at initialisation, we track the following metrics presented by Chaudhry et al. (2019) and Lopez-Paz & Ranzato (2017) :

$$\textbf{Average Accuracy : } \text{ACC} = \frac{1}{T} \sum_{\tau=1}^{T} a_{T,\tau} \tag{160}$$

$$\textbf{Backward Transfer : } \text{BWT} = \frac{1}{T-1} \sum_{\tau=1}^{T} a_{T,\tau} - a_{\tau,\tau} \tag{161}$$

$$\textbf{Forward Transfer : } \text{FWT} = \frac{1}{T-1} \sum_{\tau=1}^{T} a_{\tau-1,\tau} - \bar{b}_\tau \tag{162}$$

$$\textbf{Average Forgetting : } \text{AFM} = \frac{1}{T-1} \sum_{\tau=1}^{T-1} \max_{t \in \{1,...,T-1\}} (a_{t,\tau} - a_{T,\tau}) \tag{163}$$

### F.5 BASELINES

In addition to the standard baselines, SGD, EWC, SI and MAS, we compare OGD+ against OGD and Stable SGD, which was recently introduced by Mirzadeh et al. (2020). They show that Stable SGD outperforms OGD on all benchmarks, in order for the evaluation to be as fair and comprehensive as possible, we also include this method in our benchmarks.

#### F.5.1 COMPLEMENTARY RESULTS - CONTINUAL LEARNING METRICS

**Permuted MNIST**  Table 9 shows that OGD+ outperforms the baselines on AAC, while it remains competitive on AFM and BWT.

| Method | AAC | BWT | FWT | AFM |
|---|---|---|---|---|
| **Naive SGD** | 76.31 (±1.89) | -19.27 (±2.0) | 1.12 (±1.14) | -19.27 (±2.0) |
| **EWC** | 80.85 (±1.14) | -13.69 (±1.22) | 0.97 (±0.8) | -13.69 (±1.22) |
| **SI** | 86.69 (±0.4) | **-4.02 (±0.3)** | 2.0 (±1.1) | **-4.02 (±0.3)** |
| **MAS** | 85.96 (±0.72) | -4.84 (±0.73) | **2.51 (±1.36)** | -4.84 (±0.73) |
| **Stable SGD** | 78.17 (±0.76) | -11.63 (±1.03) | 1.02 (±0.15) | -11.63 (±1.03) |
| **OGD** | 85.0 (±0.86) | -9.71 (±0.95) | 0.19 (±0.71) | -9.71 (±0.95) |
| **OGD+** | **88.65 (±0.38)** | -5.98 (±0.36) | 0.92 (±1.84) | -5.98 (±0.36) |

Table 9: Comparison of the average accuracy, average forgetting, forward transfer and backward transfer of several methods on the Permuted MNIST benchmark.

**Rotated MNIST**   Table 10 shows that OGD+ is competitive with the baselines on AAC and AFM, while it underperforms on the other metrics. Also it draws an improvement over OGD on AAC, BWT and AFM. One probable reason is the relatively low overparameterization of the setting as shown in Table 1.

| Method | AAC | BWT | FWT | AFM |
|---|---|---|---|---|
| **Naive SGD** | 71.06 (±0.41) | -25.99 (±0.46) | **85.92 (±0.61)** | -26.37 (±0.45) |
| **EWC** | 80.96 (±0.42) | -9.03 (±0.44) | 79.01 (±0.58) | -9.58 (±0.46) |
| **SI** | 75.33 (±0.55) | -18.62 (±0.6) | 82.66 (±0.61) | -19.24 (±0.58) |
| **MAS** | 80.55 (±0.46) | -2.78 (±0.43) | 70.85 (±0.64) | -6.0 (±0.48) |
| **Stable SGD** | **88.92 (±0.19)** | **-2.22 (±0.54)** | 78.73 (±1.05) | **-2.8 (±0.53)** |
| **OGD** | 79.17 (±0.33) | -17.17 (±0.35) | 85.66 (±0.61) | -17.77 (±0.35) |
| **OGD+** | 87.73 (±0.5) | -7.88 (±0.55) | 85.45 (±0.64) | -8.57 (±0.53) |

Table 10: Comparison of the average accuracy, average forgetting, forward transfer and backward transfer of several methods on the Rotated MNIST benchmark.

**Split CIFAR100**   Table 11 shows that OGD+ is not competitive on Split CIFAR100 on all metrics. Also it draws the same performance as OGD on all metrics, one probable reason is the overparameterization of the setting as shown in Table 1.

| Method | AAC | BWT | FWT | AFM |
|---|---|---|---|---|
| **Naive SGD** | 50.77 (±3.99) | -31.63 (±4.26) | 0.31 (±1.17) | -31.82 (±4.1) |
| **EWC** | 56.82 (±1.75) | -20.32 (±2.5) | 0.56 (±1.26) | -20.43 (±2.44) |
| **SI** | 66.66 (±2.07) | -9.41 (±2.51) | **0.74 (±1.26)** | -10.03 (±2.38) |
| **MAS** | 66.33 (±1.13) | **-3.31 (±1.12)** | 0.56 (±0.93) | **-4.64 (±0.98)** |
| **Stable SGD** | **72.86 (±0.9)** | -11.14 (±0.99) | -0.19 (±1.69) | -11.14 (±0.99) |
| **OGD** | 61.82 (±1.24) | -20.84 (±1.44) | 0.33 (±1.02) | -20.86 (±1.41) |
| **OGD+** | 61.11 (±1.31) | -21.56 (±1.29) | 0.32 (±0.97) | -21.59 (±1.27) |

Table 11: Comparison of the average accuracy, average forgetting, forward transfer and backward transfer of several methods on the Split CIFAR100 benchmark.

**CUB200**   Table 12 shows that OGD+ is not competitive on the CUB200 benchmark on all metrics. Also it draws the same performance as OGD on all metrics, one probable reason is the overparameterization of the setting as shown in Table 1.

| Method | AAC | BWT | FWT | AFM |
|---|---|---|---|---|
| **Naive SGD** | 53.93 (±0.86) | -17.07 (±0.88) | **0.15 (±0.45)** | -17.07 (±0.88) |
| **EWC** | 62.15 (±0.37) | -3.88 (±0.49) | -0.12 (±0.57) | -3.88 (±0.49) |
| **SI** | **63.17 (±0.42)** | **-3.24 (±0.63)** | -0.23 (±0.51) | **-3.43 (±0.46)** |
| **MAS** | 61.43 (±0.68) | -7.76 (±0.87) | -0.17 (±0.8) | -7.85 (±0.9) |
| **Stable SGD** | 58.79 (±0.36) | -6.16 (±0.23) | -0.01 (±0.8) | -6.23 (±0.17) |
| **OGD** | 57.89 (±0.89) | -12.97 (±0.94) | 0.02 (±0.55) | -12.99 (±0.92) |
| **OGD+** | 57.99 (±0.52) | -12.88 (±0.51) | 0.08 (±0.53) | -12.89 (±0.51) |

Table 12: Comparison of the average accuracy, average forgetting, forward transfer and backward transfer of several methods on the CUB200 benchmark.

F.5.2   COMPLEMENTARY RESULTS - VALIDATION ACCURACY THROUGH TRAINING

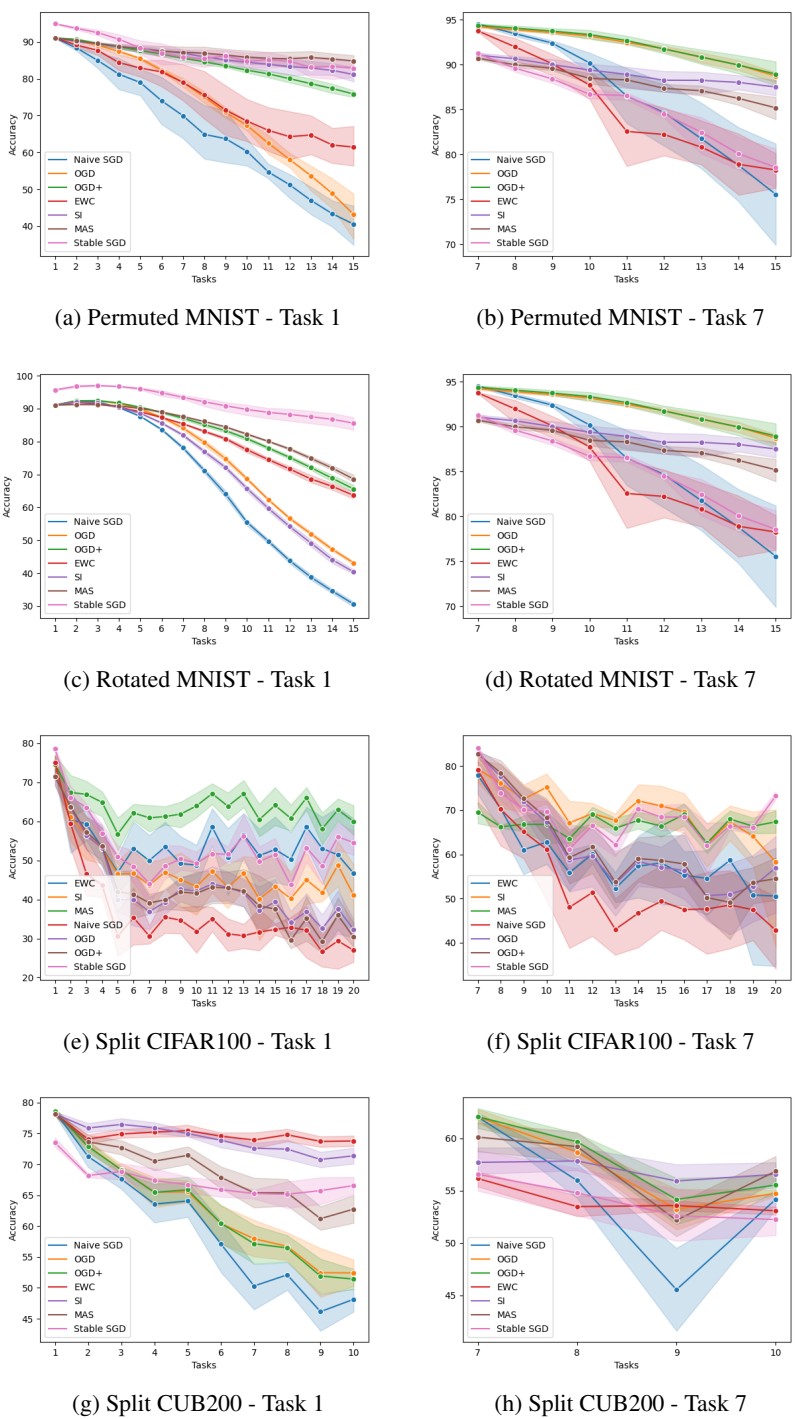

(a) Permuted MNIST - Task 1

(b) Permuted MNIST - Task 7

(c) Rotated MNIST - Task 1

(d) Rotated MNIST - Task 7

(e) Split CIFAR100 - Task 1

(f) Split CIFAR100 - Task 7

(g) Split CUB200 - Task 1

(h) Split CUB200 - Task 7

Figure 6: Test accuracy through tasks for different Continual Learning methods on the MNIST and CIFAR100 and CUB200 benchmarks. The y-axis is truncated for clarity. We report the mean and standard deviation over 5 independent runs. The test error is measured at the end of each task.

