# OpenReview forum: "Generalisation Guarantees For Continual Learning With Orthogonal Gradient Descent"
_ICLR.cc/2021/Conference — Reject_

### Official Review · AnonReviewer3 · 2020-10-27
**Generalisation Guarantees For Continual Learning With Orthogonal Gradient Descent**

**Rating:** 5
**Confidence:** 3

**Review:**

##########################################################################

Summary:


This paper studies the theoretical aspect of a continual learning method called orthogonal gradient descent (OGD).
In this study, authors leverage Neural Tangent Kernel and over parameterized neural networks to prove the generalization of OGD.

##########################################################################

Reasons for score:


Overall, I vote for rejection. I like the idea of the paper to analyze an exist method from different aspect and even improving it. However, my major concern is about the clarity of the paper (see cons below).


##########################################################################Pros:


1. The paper investigate an important problem in continual learning framework which is the generalization.


2. This paper provides some experiments to show the effectiveness of the proposed framework.


##########################################################################

Cons:

1. Unfortunately, the paper is not clear and very difficult to follow.
a) For instance, NTK is referred without explaining it well first.
In page 2, authors use "CL" for referring to continual learning but it has not been defined.
b) There are many typos and capital letters have been used inappropriately.
c) f_t has not been defined.
2- Although the proposed method provides several experiments, there are still many other methods and datasets that have been ignored. There has been recent studies and frameworks that have outperformed OGD, it woud be great if you include them in your baselines. For instance, SGD+Droput in https://arxiv.org/pdf/2004.11545.pdf beats OGD.

3-  There are many metrics to evaluate continual learning frameworks like backward transfer(BWT) or average accuracy over tasks.
I would suggest the authors to look at the defined metrics in GEM (gradient episodic memory), https://arxiv.org/abs/1706.08840, and compute those values.


##########################################################################

Questions during rebuttal period:


- Please address and clarify the cons above .
- Would you please elaborate more what could be the superior performance of OGD+ on Rotated Mnist dataset w.r.t OGD?
- What is the time complexity of OGD+?

---

> ### Author Response · Authors · 2020-11-19
> **Thank you for your comments and suggestions**
>
> Thank you for your appreciation and comments  Please find our responses below.
>
> >“a) For instance, NTK is referred without explaining it well first. In page 2, authors use "CL" for referring to continual learning but it has not been defined. b) There are many typos and capital letters have been used inappropriately. c) f_t has not been defined. “
>
> We apologise for the confusion incurred by the lack of clarity.
> We took due note of your suggestions and we have revised and updated the flow of the paper to ensure the notions are introduced clearly.
>
> >“2- Although the proposed method provides several experiments, there are still many other methods and datasets that have been ignored. There has been recent studies and frameworks that have outperformed OGD, it woud be great if you include them in your baselines. For instance, SGD+Dropout in https://arxiv.org/pdf/2004.11545.pdf beats OGD.”
>
> Thanks for your suggestions and for the reference.
>
> **Comparison with other methods and datasets**
>
> In order to provide a broader picture of the performance of OGD+, we have added new results for the baselines SGD+Dropout, EWC, SI and MAS. (Sec. 7.3 and App. F.4, pages 35 to 38)
> We have also added results for the CUB200 dataset [1] on these baselines, except SGD+Dropout for which we are still running experiments forCUB200.
>
> **Ablation study : OGD and OGD+**
>
> Also, we wanted to clarify the main reason we didn’t include the other Continual Learning baselines in our experiments, and restricted the comparison to OGD and OGD+.
> The introduction of OGD+ was intended for an ablation study in order to verify the applicability of the theoretical framework in practice.
>
> Our analysis relies on the overparameterization assumption, which implies that the Jacobian is constant through tasks. Theorem 2 follows from this property. However, in practice, this assumption may not hold and forgetting is observed for OGD. We study the impact of the variation of the Jacobian in practice on OGD.
> In order to measure this impact, we compare the performance of OGD, which doesn’t take into account the variation of the Jacobian to OGD+ which does. As opposed to OGD, OGD+ takes into account the Jacobian’s variation by updating all the stored Jacobians at the end of each task.
>
> Table 1 (Sec 7.2) shows that on the least overparameterized benchmarks, OGD+ presents more robustness, while there is no significant difference on the most overparameterized benchmarks.
>
> This results highlights the scope of applicability of our analysis, and suggests that an important next step is extending the theoretical framework to the non-overparameterized setting, in order to capture the training dynamics of the MNIST benchmarks for instance.
>
> This result also shows that OGD is expected to perform better in overparameterised settings, and that accounting for the Jacobian's variation is a well motivated improvement in non-overparameterised settings.
>
> **Clarity**
>
> We apologise for the lack of clarity, we have written over the whole experiments section for more clarity.
>
>
> > 3- There are many metrics to evaluate continual learning frameworks like backward transfer(BWT) or average accuracy over tasks. I would suggest the authors to look at the defined metrics in GEM (gradient episodic memory), https://arxiv.org/abs/1706.08840, and compute those values.
>
> Thanks for your suggestion.
> In our new experiments, we have integrated the metrics Average Accuracy (AAC), Forward Transfer (FWT), Backward Transfer (BWT) and Average Forgetting Measure (AFM) as suggested.
> We report concise results in the main paper (Sec. 7.2 and 7.3).
> We also report the full results on all metrics in App. F.5.1 (pages 35 to 36).
>
> **References**
> [1] Jung et al., Continual Learning with Node-Importance based Adaptive Group Sparse Regularization (https://arxiv.org/abs/2003.13726)

---

> > ### Author Response · Authors · 2020-11-19
> > **Thank you for your comments and suggestions - Continued**
> >
> > >“What is the time complexity of OGD+?”
> >
> > The time complexity of OGD+ is the same as OGD, the only difference is updating the Jacobians memory, which implies a backpropagation on all samples in the memory at the end of each task.
> > Denoting :
> > - T : the number of tasks
> > - M : the memory size
> > - N : the number of samples per task
> >
> > Considering that the backprop cost is constant, the total complexity of the memory update operation is O(T^2 M^2)
> >
> > The time complexity of OGD is : O(NT) + O(M^2T)
> >
> > Therefore the complexity of OGD+ is O(NT) + O(M^2T) + O(T^2 M^2)

---

### Official Review · AnonReviewer2 · 2020-10-27
**Novel Theory for Continual Learning in the context of Orthogonal Gradient Descent.**

**Rating:** 6
**Confidence:** 3

**Review:**

The authors use a Neural Tangent Kernel (NTK) approximation of wide neural nets to establish generalization bounds for continual learning (CL) using stochastic gradient descent (SGD) and orthogonal gradient descent (OGD).  In this regime, the authors prove that OGD does not suffer from catastrophic forgetting of training data.  The authors additionally introduce a modification to OGD which causes significant performance improvements in the Rotated MNIST and Permuted MNIST problems.  OGD involves storing feature maps from data points from previous tasks.  The modified OGD method (OGD+) additionally stores feature maps from the current task.

The primary contribution of this paper is the theoretical analysis of continual learning.  Given that the CL problem does not have an extensive theoretical foundation, the generalization bound in this paper is a notable advance. The theory presented also provides a justification for the empirical observations observed by the authors that as overparameterization increases, the effect of catastrophic forgetting decreases in a variety of CL task setups.  The primary drawback of the paper is that the authors do not compare the OGD+ algorithm to other continual learning algorithms (synaptic intelligence, elastic weight consolidation, etc.).  As a result it is difficult to know how OGD+ compares to alternatives.  It is not clear to the reviewer why improving OGD to OGD+ is itself a contribution.  Given the expense occurred by OGD-type methods in storing ever increasing numbers of directions, it would be important to know the comparison of this method with others.

Minor comments:

(1) Section 3.2: f^* is not defined as of this point in the paper.
(2) Theorem 1: The theorem needs a quantifier of lambda
(3) Line above Remark 1 k_\tau -> \kappa_\tau
(4) Theorem 2: The paper should define what "is in the memory" means when introducing OGD
s
(5) Theorem 3: Definition of R_T has incorrect dummy index in the summation

---

> ### Author Response · Authors · 2020-11-19
> **Thanks for your appreciation and comments**
>
> Thanks for your appreciation, we address your comment below.
>
> > “The primary drawback of the paper is that the authors do not compare the OGD+ algorithm to other continual learning algorithms (synaptic intelligence, elastic weight consolidation, etc.). As a result it is difficult to know how OGD+ compares to alternatives. It is not clear to the reviewer why improving OGD to OGD+ is itself a contribution. Given the expense occurred by OGD-type methods in storing ever increasing numbers of directions, it would be important to know the comparison of this method with others.”
>
> Thanks for your feedback.
>
> **Comparing OGD+ to other methods :**
>
> We have added new results for the baselines SGD+Dropout, EWC, SI and MAS. (Sec. 7.3 and App. F.4, pages 35 to 38)
> We have also added results for the CUB200 dataset [1] on these baselines, except SGD+Dropout for which we are still running experiments forCUB200.
>
> **Is OGD+ a contribution ?**
>
> We mainly leverage OGD+ as an ablation of OGD, in order to highlight the limits of the theoretical framework in non-overparameterized settings.
> The goal behind the OGD+ experiment is to study, in non-overparameterized settings in practice, the importance of accounting for the Jacobian’s variation for OGD. The motivation behind this experiment is that Theorem 2, which states the robustness of OGD to CF, relies on the assumption that the Jacobian is constant (overparameterization), which may not hold in practice.
>
> - Table 1 shows that accounting for the Jacobian’s variation implies a significant improvement on non-overparameterized benchmarks, indicating that the assumption may be too strong for the MNIST settings, and a more refined analysis may be needed in order to explain more precisely the properties of OGD in these settings, which are not under the NTK regime.
> This result also shows that OGD is expected to work better on overparameterized benchmarks.
> - Regarding the performance of OGD+ in comparison with other baselines, Table 2 shows that OGD+ is competitive with the Continual Learning baselines on non-overparameterized benchmarks, for which it draws a significant improvement over OGD, by taking into account the variation of the Jacobian which is a property of these benchmarks.
>
> **Clarity**
>
> We have written over the whole experiments section (Sec. 7) for clarity. We also highlight more clearly the ablation study, its motivation and implications.
>
> >“Minor comments
> (1) Section 3.2: f^* is not defined as of this point in the paper. (2) Theorem 1: The theorem needs a quantifier of lambda (3) Line above Remark 1 k_\tau -> \kappa_\tau (4) Theorem 2: The paper should define what "is in the memory" means when introducing OGD s (5) Theorem 3: Definition of R_T has incorrect dummy index in the summation”
>
> Thank you for the additional suggestions. We took due note of them and applied the corrections.
>
> **References**
>
> [1] Jung et al., Continual Learning with Node-Importance based Adaptive Group Sparse Regularization (https://arxiv.org/abs/2003.13726)

---

### Official Review · AnonReviewer1 · 2020-10-28
**Theoretical analysis for OGD for continual learning**

**Rating:** 5
**Confidence:** 4

**Review:**

The paper provides a theoretical analysis on the OGD based continual learning method. The method is in fact proposed by a previous paper (Farajtabar et al. 2019) and the current paper shows a generalization bound for the regression case. The result (Thm 3) compares the generalization bounds between SGD and OGD and shows OGD leads to a tighter bound. The theorem is also based on the bound on the Rademacher Complexity (Lemma 1).  The paper also suggests OGD+, which stores some data points from past tasks. They also present some experimental results on small benchmark datasets, and show OGD+ outperforms SGD and OGD.

While the paper makes an interesting attempt on theoretical analyses of OGD based continual learning method, I feel the result is quite limited only to the OGD scheme. Also, the result is for regression, as shown in the loss function in Sec 3.2, but the experiments are on classification, so it's not clear with the connection with the theory and the experiments. The results also seem to be somewhat simple derivations from the known papers, like Jacot et al., (2018) and Liu et al, (2019).

The experimental results are also very limited  and weak since it only compares with SGD, an obvious weak scheme that suffers from catastrophic forgetting, and does not compare with any other continual learning baselines. For example, the state-of-the-art on CIFAR-100 is around 65%, and the performance of OGD is very weak. Even though the paper aims for a theoretical contribution, it is very limited only for OGD based scheme, which is not strong in practice. So, I am not sure about the significance of the contribution of the paper. But, I haven't fully read the entire proof of the paper, and I may have missed some details regarding the proof. I would like to see other reviewers' opinion as well.

What about comparing with more enlarged and various benchmark datasets beyond MNIST and CIFAR-100, like CUB200 or Omniglot as shown in https://arxiv.org/pdf/2003.13726.pdf ? How does OGD or OGD+ compares with other baselines like EWC or MAS?

---

> ### Author Response · Authors · 2020-11-19
> **Thank you for your suggestions and comments**
>
> Thank you for your comments and suggestions. We address your comments below.
>
> >“Also, the result is for regression, as shown in the loss function in Sec 3.2, but the experiments are on classification, so it's not clear with the connection with the theory and the experiments.”
>
> **The analysis is for regression only :**
>
> We agree on the importance of a theoretical framework for the classification setting, however this setting brings additional challenges due to the non linearity of the classification loss function, which makes the analysis intractable.
> Even though our analysis is for the regression setting only, it provides multiple insights on the foundations of Continual Learning, such as Theorem 1, which describes the transfer of knowledge across tasks, or Theorem 2, which states the robustness of OGD to Catastrophic Forgetting.
> Also, our experiments in Sec. 7.1 and Sec. 7.2 concur with our theoretical analysis, even though it is limited to the regression setting.
>
> **Connection between the theory and the experiments :**
>
> Classification experiments highlight the similarity to the regression analysis in Continual Learning as follows :
> - Experiment 1 (7.1) checks the validity of Theorem 2 in the classification setting. Figure 1 shows that Catastrophic Forgetting decreases with overparameterization which concurs with Theorem 2 states that there is no Catastrophic Forgetting in the overparameterized regression case.
> - Experiment 2 (7.2) checks the applicability of the theoretical framework in practice for non overparameterized settings.
> Table 1 shows that accounting for the Jacobian’s variation in non overparameterized settings is in part responsible for the failure of OGD to Catastrophic Forgetting in practice. This results enforces Theorem 2 and  indicates that the constant Jacobian assumption is critical for a further theoretical investigation of the OGD algorithm in non-overparameterized settings.
>
>
> >“The results also seem to be somewhat simple derivations from the known papers, like Jacot et al., (2018) and Liu et al, (2019).”
>
> We respectfully disagree with the reviewer on this point, the proof techniques and mathematical tools we use are different from the mentioned works as follows :
> - Jacot et al., (2018) used extensively the Kernel Tangent mathematical object, while in our case we tackled the problem without relying on this object. Instead, we build the analysis on a linear approximation which makes the analysis more tractable in our setting. We found that the mathematical tools used by Jacot et al., (2018), even though they were powerful, were more challenging to use to build the framework for Continual Learning.
> - Liu et al, (2019) focused on the two-layer RELU network setting and relied on Gaussian approximations in order to make the analysis tractable. In our analysis, we did not manipulate the distributions of the neural network weights.
> - However, we used multiple proof techniques and mathematical tools presented by Hu et al. (2020) (https://arxiv.org/abs/1905.11368). In this work, they studied noisy supervision in the NTK framework.
>
> Also, the analysis for the OGD Continual Learning setting brings additional challenges which are specific to Continual Learning and OGD such as :
> - All the proofs are for the OGD optimisation algorithm, while to our knowledge most papers on the Neural Tangent Kernel study the SGD optimisation algorithm
> - The proof of Theorem 1 (Continual Learning as a recursive Kernel Regression)
> - The proof of Lemma 1 by extending the standard bound to the Continual Learning setting (Rademacher complexity for Continual Learning)
> - The proof of Theorem 2 (Non-forgetting property of OGD)

---

> > ### Author Response · Authors · 2020-11-19
> > **Thank you for your suggestions and comments - Continued**
> >
> > >“What about comparing with more enlarged and various benchmark datasets beyond MNIST and CIFAR-100, like CUB200 or Omniglot as shown in https://arxiv.org/pdf/2003.13726.pdf ? How does OGD or OGD+ compare with other baselines like EWC or MAS?”
> >
> > >“ The experimental results are also very limited and weak since it only compares with SGD, an obvious weak scheme that suffers from catastrophic forgetting, and does not compare with any other continual learning baselines. “
> >
> > Thanks for your suggestions and for the reference.
> >
> > **Comparing against more benchmarks and methods**
> > We have added new results for the baselines SGD+Dropout, EWC, SI and MAS. (Sec. 7.3 and App. F.4, pages 35 to 38)
> > We have also added results for the CUB200 dataset [1] on these baselines, except SGD+Dropout for which we are still running experiments for CUB200.
> >
> > **Ablation study : OGD and OGD+**
> > Also, we wanted to clarify the main reason we did not include the other Continual Learning baselines in our experiments, and restricted the comparison between OGD and OGD+.
> >
> > OGD+ was intended as an ablation of OGD, in order to highlight the limits of the theoretical framework in non-overparameterized settings.
> >
> > The goal behind the OGD+ experiment is to study, in non-overparameterized settings in practice, the importance of accounting for the Jacobian’s variation for OGD. The motivation behind this experiment is that Theorem 2, which states the robustness of OGD to CF, relies on the assumption that the Jacobian is constant (overparameterization), which may not hold in practice.
> > - Table 1 shows that accounting for the Jacobian’s variation implies a significant improvement on non-overparameterized benchmarks, indicating that the assumption may be too strong for the MNIST settings in practice, and a more refined analysis may be needed in order to explain more precisely the properties of OGD in non-NTK regime settings.
> > - Regarding the performance of OGD+ in comparison with other baselines, Table 2 shows that OGD+ is competitive with the Continual Learning baselines on non-overparameterized benchmarks, for which it draws a significant improvement over OGD, by taking into account the variation of the Jacobian which is a property of these benchmarks.
> >
> > **Clarity**
> > We have written over the whole experiments section (Sec. 7) for clarity. We also highlight more clearly the ablation study, its motivation and implications.
> >
> > >“Even though the paper aims for a theoretical contribution, it is very limited only for OGD based scheme, which is not strong in practice. So, I am not sure about the significance of the contribution of the paper.”
> >
> > While we agree that the analysis is limited to the OGD based scheme, to our knowledge, it is one of the first theoretical works on convergence and optimisation for Continual Learning in the literature, which may provide keys for further investigation.
> > Also, our work is motivated by the following :
> > - understanding the theoretical properties of the OGD algorithm may lead to insights to improve it or to design more robust algorithms.
> > - providing mathematical tools and proof techniques to analyse other Continual Learning algorithms.
> >
> > In Appendix A, we present early theoretical results extending the presented framework in order to draw a theoretical connection between the OGD and the A-GEM algorithms. This early result indicates that the theoretical framework may lead to new viewpoints on existing Continual Learning algorithms beyond OGD.
> >
> > **References**
> > [1] Jung et al., Continual Learning with Node-Importance based Adaptive Group Sparse Regularization (https://arxiv.org/abs/2003.13726)

---

### Author Response · Authors · 2020-11-22
**We have completed the experiments - Overview of the updates**

Thanks again for your insightful suggestions and comments.

We have uploaded a new revision, which comprises all the results.

Additionally, for clarity, we wanted to present a brief overview of the changes we have applied to the manuscript overall :
- **Experiments** :
   - Added the benchmark against the baselines EWC, MAS, SI and Stable SGD [1] and the dataset CUB200 [2] (Sec. 7, Table 2)
   - Reported the metrics Average Forgetting, Average Accuracy, Forward Transfer and Backward Transfer for the benchmark  (Tables 1 and 2 (p. 9), and Tables 9 to 12 (p. 36 to 37))
   - Reported the overparameterization ratio of the benchmarks (Table 1)
   - Rewrote the experiments section (Sec. 7 (p. 7 to 9) ) and the experiments appendix (App. F (p. 30 to 37) ) following the updates above and the other feedback

- **Clarity** :
   - Updated the introduction in order to introduce the concepts more clearly
   - Fixed and clarified the preliminaries for more clarity
   - Rewrote the experiments section highlighting more clearly the ablations and the goal behind the experiments
   - Fixed some small mistakes and typos

Please don't hesitate if you have additional concerns, comments or suggestions.

Thanks again.

**References**
- [1] Mirzadeh et al., Understanding the Role of Training Regimes in Continual Learning (https://arxiv.org/abs/2006.06958)
- [2] Jung et al., Continual Learning with Node-Importance based Adaptive Group Sparse Regularization (https://arxiv.org/abs/2003.13726)

---

### Decision · Program_Chairs · 2021-01-07
**Final Decision**

**Decision:**

Reject

**Comment:**

The reviewers were excited by the paper's theoretical contribution to continual learning, since that aspect of continual learning is underdeveloped.  However, all reviewers (including the most positive reviewer during discussions) expressed that the paper would benefit from revisions to improve the clarity and the thoroughness of comparisons in the paper.  The paper's focus on OGD is not necessarily an issue for it to be of use to the community, as mentioned as a negative point in one review that other reviewers disagreed with. The authors are encouraged to revise this paper incorporating the reviewers' suggestions.